# PARAMETER-EFFICIENT FINE-TUNING DESIGN SPACES

**Jiaao Chen**[†,*] **Aston Zhang**[‡]**, Xingjian Shi**[‡]**, Mu Li**[‡]**, Alex Smola**[‡]**, Diyi Yang**[◇]
[†]Georgia Institute of Technology, [‡]Amazon Web Services, [◇]Stanford University

## ABSTRACT

The aim of parameter-efficient fine-tuning is to achieve performance that is comparable to fine-tuning, but with fewer trainable parameters. Several hand-crafted strategies, such as Adapters, Prefix Tuning, BitFit, and LoRA, have been proposed, but it remains unclear whether there are underlying design patterns. Thus, we present a parameter-efficient design paradigm and identify design patterns that are applicable to various experimental settings. Instead of developing another individual tuning strategy, we introduce design spaces that parameterize tuning structures and strategies. These design spaces consist of four components: layer grouping, trainable parameter allocation, tunable groups, and strategy assignment. Our experiments reveal the following design patterns: (i) group layers in a spindle pattern, (ii) allocate trainable parameters evenly among layers, (iii) tune all groups, and (iv) assign appropriate tuning strategies to each group. These patterns lead to new methods for parameter-efficient fine-tuning, which we show experimentally outperform existing strategies across various backbone models and NLP tasks[1].

## 1 INTRODUCTION

Large pre-trained models have shown to achieve state-of-the-art results in many downstream natural language processing tasks, by fine-tuning on task-specific labeled data (Devlin et al., 2019; Liu et al., 2019; Yang et al., 2019; Joshi et al., 2019; Sun et al., 2019; Clark et al., 2019; Lewis et al., 2020a; Bao et al., 2020; He et al., 2020; Raffel et al., 2020; Ziems et al., 2022). However, the cost of fine-tuning all parameters and storing them separately for each task is high in terms of computational and storage resources, e.g., 355 million parameters for RoBERTa (Liu et al., 2019) and 175 billion parameters for GPT-3 (Brown et al., 2020). This makes it challenging to deploy in real-world natural language processing (NLP) systems that handle multiple tasks.

To make pretrained models more efficient for specific downstream tasks, various strategies have been proposed that only learn a small number of extra parameters while keeping the rest frozen (Houlsby et al., 2019b; Pfeiffer et al., 2021; Li & Liang, 2021; Brown et al., 2020; Lester et al., 2021b; Schick & Schütze, 2021; Ziems et al., 2022). One such strategy is adapter tuning (Houlsby et al., 2019b), which adds small neural modules (adapters) to each layer of the pretrained network, and only trains the adapters during fine-tuning. Other methods, such as prefix tuning (Li & Liang, 2021) and prompt tuning (Lester et al., 2021a), have been inspired by the success of controlling pretrained models through textual prompts (Brown et al., 2020). These methods prepend tunable tokens to the input or hidden layers, and only train these tokens during fine-tuning. BitFit (Zaken et al., 2021) updates the bias terms of pretrained models while freezing the rest, while LoRA (Hu et al., 2021) decomposes attention weight gradients into low-rank matrices to reduce the number of trainable parameters. He et al. (2022) proposed a unified view of these strategies, illustrating their differences and connections, but like its predecessors, the method is still equally applied to different layers of the pretrained network.

Most current fine-tuning strategies to adapt pretrained models to specific tasks are effective, but they are often developed through manual design processes without considering potential design patterns

---

[*]Work done during an internship at Amazon Web Services. Correspondence to Jiaao Chen <jiaaochen@gatech.edu> and Aston Zhang <astonz@amazon.com>.

[1]We will release our code at `https://github.com/amazon-science/peft-design-spaces`.

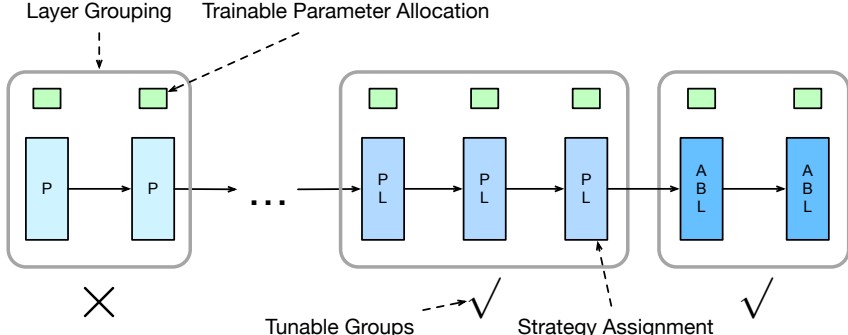

Figure 1: The design space is characterized by: (i) Grouping of consecutive layers, (ii) The allocation of the number of trainable parameters to each layer, (iii) The selection of groups that will be fine-tuned, and (iv) The assignment of appropriate strategies, such as Adapter (A), Prefix (P), BitFit (B), or LoRA (L), to each group.

across these strategies, different backbone models, and downstream tasks. The effectiveness of different strategies is also unclear as they are usually applied separately, and it's unknown how they reinforce or complement each other (Mao et al., 2022). Our aim is to gain a comprehensive understanding of the fine-tuning design and uncover interpretable and widely applicable design patterns.

Instead of creating yet another strategy to be applied uniformly to various pretrained layers, we present *parameter-efficient fine-tuning design spaces* that allow customization of both tuning structures and strategies. These design spaces are comprised of four main components, as illustrated in Figure 1: *layer grouping*, *trainable parameter allocation*, *tunable groups*, and *strategy assignment*.

We start our journey towards parameter-efficient fine-tuning design using a relatively unconstrained design space. We then narrow this space through successive rounds of comparison, using random sampling and while enforcing constraints such as equal layer grouping. Through this process, we discover several key design patterns, including layer grouping in a spindle pattern, uniform allocation of trainable parameters, tuning all groups, and appropriate strategy assignments. Our new methods outperform existing parameter-efficient fine-tuning strategies. We demonstrate the effectiveness of our approach using T5 (Raffel et al., 2020) and classification tasks, but find that the discovered design patterns are applicable to other backbones (such as RoBERTa (Liu et al., 2019), BART (Lewis et al., 2020b) and XLNet (Yang et al., 2019)), and NLP tasks (e.g., summarization, machine translation, and eight SuperGLUE datasets).

Our contributions are: (i) The introduction of parameter-efficient fine-tuning design spaces. (ii) The discovery of several design patterns in parameter-efficient fine-tuning through comprehensive experiments. (iii) The creation of parameter-efficient fine-tuning methods based on the discovered design patterns, which outperform existing strategies on various backbone models and NLP tasks.

## 2 RELATED WORK

Our work is closely related to and builds on work about network design spaces and parameter-efficient fine-tuning. We discuss the connections and differences below.

**Network Design Spaces.** Many works designed neural network models via an ad-hoc discovery of new design choices that improve performance (Radosavovic et al., 2019), such as the use of deeper architectures or residual connections. Recent work (Radosavovic et al., 2020; You et al., 2020; Radosavovic et al., 2019) focuses on the design space to discover new design principles for convolutional neural networks (Radosavovic et al., 2020) and graph neural networks (You et al., 2020). Inspired by this work we focus on the design spaces to rethink parameter-efficient fine-tuning, with the goal of discovering design patterns that are applicable to different settings.

**Parameter-Efficient Fine-Tuning for NLP.** As pretrained models increase in size, storing and fine-tuning them becomes increasingly expensive and unfeasible for those without ample computational

resources. A growing body of research is aimed at finding alternatives to fine-tuning large-scale models that reduce memory and storage costs. Some researchers have proposed using bottleneck layers with skip-connections to adapt large models, as seen in works such as Houlsby et al. (2019a), (Stickland & Murray, 2019), (Pfeiffer et al., 2020), and (Rebuffi et al., 2017). Other works focus on identifying and training only a subset of all model parameters, such as (Zhao et al., 2020) and (Guo et al., 2020). More recent research explores low-rank decomposition (Zhang et al., 2021) and the injection of trainable low-rank decomposition matrices into each layer (Hu et al., 2021; Karimi Mahabadi et al., 2021).

Li & Liang (2021) introduced prefix-tuning, where a set of prefixes is added to autoregressive language models or both encoders and decoders, while Lester et al. (2021b) proposed adding virtual tokens to the embedding layer. Another approach, side-tuning, was introduced in (Sung et al., 2022). He et al. (2022) and Ding et al. (2022). They proposed a unified view of existing parameter-efficient fine-tuning strategies. In yet another approach, Mao et al. (2022) introduced a unified framework to combine various methods through mixture-of-experts.

Our research focuses on the general *design spaces* of parameter-efficient fine-tuning, providing a more comprehensive view of this method. By experimenting and refining design spaces, we aim to discover design patterns for parameter-efficient fine-tuning.

## 3    COMPONENTS OF DESIGN SPACES

Our goal is not to list all possible design choices, but to show how design spaces can guide parameter-efficient fine-tuning research. As such, we pick a representative subset for each of the following four components: (i) layer grouping, (ii) trainable parameter allocation, (iii) tunable groups, and (iv) strategy assignment. Figure 1 provides an example.

Given these choices, we sample from a distribution over them, then pick a subset that performs the best, narrowing down the set of choices. Given that more restrictive set, we repeat the procedure by sampling and picking a now even more restrictive subset until we arrive at a concise description of the design space. Quite understandably, this is very costly when dealing with Large Language Models. Taking a leaf out of (Radosavovic et al., 2020) we perform our experiments using a sufficiently cheap model, in this case T5-base and T5-3b, unless stated otherwise. Further details will be discussed in the next section. For now let's review the set of choices available.

**Layer Grouping.** Different layers in pre-trained models capture varying information and behave differently. For example, the authors of (Jawahar et al., 2019) found that the $3, 4, 5, 6, 7, 9, 12$th layers have the most representation power in BERT and each layer captures a different type of information, ranging from surface to syntactic to semantic level representation of text. For instance, the 9th layer performs well in semantic tasks such as checking random swaps of coordinated clauses, while the 3rd layer is best suited for surface tasks like predicting sentence length.

When adapting these pre-trained models for downstream tasks, it's crucial to group layers with similar behaviors together. This is critical to the design and proper implementation of parameter-efficient fine-tuning strategies. In this design component, we study patterns of how to group consecutive layers in pre-trained models (e.g., transformer layers in T5) during the fine-tuning process.

**Trainable Parameter Allocation.** In parameter-efficient fine-tuning, the total number of trainable parameters is usually set to a small portion of the total parameters in the pretrained model. Our study will explore different ways to allocate the predefined number of trainable parameters to the layers.

**Tunable Groups.** Not all the parameters of a pretrained model need to be updated during fine-tuning for downstream tasks. For example, BitFit (Zaken et al., 2021) only updates the bias parameters while freezing the rest. As a result, we explore which groups of parameters need to be learned during parameter-efficient fine-tuning to achieve better performance.

**Strategy Assignment.** In order to improve the parameter efficiency, different sets of strategies (Li & Liang, 2021; Lester et al., 2021b; Houlsby et al., 2019b; Hu et al., 2021) were proposed, where only a small number of (extra) parameters are tuned and the remaining parameters in these pretrained models are frozen to adapt their general knowledge to specific down-stream tasks. We hypothesize that different groups might benefit from different proper strategies (or combinations) for capturing

different types of information. More formally, given a set of individual strategies $\mathcal{A}$ for assignment, for any group $G_i$, assign a subset $\mathcal{U}_i \subset \mathcal{A}$ to each layer in $G_i$.

# 4 DISCOVERING DESIGN PATTERNS

Each design space, denoted as $\mathcal{S}_i$, consists of a set of models ($\mathcal{S}_i$-models) that satisfy the constraints characterizing the space with respect to layer grouping, trainable parameter allocation, tunable groups, and strategy assignment. To discover design patterns, we start from a relatively unconstrained parameter-efficient fine-tuning design space $\mathcal{S}_0$. We progressively refine it via $\mathcal{S}_1, \ldots \mathcal{S}_4$ by comparing the overall quality of models in design spaces enforced with different constraints (e.g., each group has the same number of layers). To quantify the overall quality of models in any design space $\mathcal{S}_i$ with a low-compute, low-epoch regime (Radosavovic et al., 2020), we randomly sample 100 models from $\mathcal{S}_i$, fine-tune with only 3 epochs [2], and compute the average of the GLUE average performance. Using such a low number of epochs is sufficient to obtain a sufficiently *representative* score to draw consistent conclusions (see Table 7 in the Appendix) that extend to a full training run.

We emphasize that our goal is to demonstrate how the perspective of design spaces can help inform parameter-efficient fine-tuning research, rather than to find out the "best" design space or method. For computational efficiency, it is beyond the scope of this work to enumerate all possible constraints with respect to the design space components (Section 3). For efficiency, we use T5-base (pretrained backbone model) as it's both representative and also sufficiently small to make experimentation with many options computationally affordable.

In this work, we follow the discovery sequence of "grouping patterns – trainable parameter allocation – tunable groups – strategy assignment": (1) To explore and understand the design patterns in all the layers in large pre-trained models in scale, it is necessary and more efficient to study the layers in the unit of groups. So we start with the grouping patterns. (2) Once figuring out the optimal grouping patterns, it is then important to explore how to allocate the trainable parameters to these different groups in order to study more subtle designs with fair comparisons (e.g., this would allow comparing different patterns of strategy assignments without the impact from different trainable parameters.). (3) Next, it becomes influential to examine which groups need to be learned during fine-tuning before we dig into the strategy assignment patterns. Because it is only meaningful to study assigning strategies to different groups after we figure out which groups need to be learned. (4) Finally, we study the tuning strategy assignment, which is the most subtle design.

## 4.1 $\mathcal{S}_0$ — THE INITIAL DESIGN SPACE

The initial relatively unconstrained design space $\mathcal{S}_0$ consists of all models without constraints on the design space components. Individual parameter-efficient fine-tuning strategies consist of Adapter, Prefix, BitFit, and LoRA. Specifically, without grouping constraints, each layer of the pretrained layer has a probability of $0.5$ to be tuned. If tuned, a random strategy, or combinations thereof, with a random amount of trainable parameters are assigned to that layer.

Before comparing more subtle design patterns such as to which tuning strategy among Adapter, Prefix, BitFit, and LoRA to pick, we begin by exploring how to *group* layers and how to allocate the total number of trainable parameters to layers.

## 4.2 $\mathcal{S}_1$ — APPLYING GROUPING CONSTRAINTS

Transformers are quite deep by now. This makes it impractical to pick a different tuning strategy for each layer. As such, the first question to ask is how to assemble the layers into groups that will be tuned using the same strategy. Inspired by Radosavovic et al. (2020), we consider *4* groups, $G_1, \ldots, G_4$, in the order of forward pass, in the experiments [3] Denote by $N_i$ the number of layers in $G_i$. As illustrated in Figure 2, we compare the following layer grouping patterns:

**Increasing** ($N_{i+1} > N_i$): the number of layers in groups gradually increases;

---

[2] We set the low epoch by observing whether it is enough for models to obtain stable performances to draw consistent conclusions (See Table 7 in the Appendix).

[3] The experimental results with 8 groups are shown in the Table 16 in the Appendix.

Table 1: Average performance (low-compute, low-epoch regime: 100 random models, 3 tuning epochs) on the GLUE datasets using the T5-base pretrained backbone. We compare adding different layer grouping constraints to the $\mathcal{S}_0$ design space.

| Layer Grouping | SST-2 | MNLI | QNLI | QQP | RTE | STS-B | MRPC | CoLA | Avg |
|---|---|---|---|---|---|---|---|---|---|
| $\mathcal{S}_0$-models | 76.9 | 70.1 | 72.5 | 73.3 | 63.6 | 71.7 | 73.8 | 24.3 | 65.7 |
| Increasing | 85.3 | 74.9 | 77.2 | 77.5 | 66.8 | 76.2 | 76.0 | 33.0 | 70.8 |
| Uniform | 84.8 | 73.7 | 78.1 | 78.6 | 68.5 | 77.8 | 79.2 | 36.1 | 72.1 |
| Decreasing | 81.9 | 72.1 | 78.3 | 76.7 | 67.3 | 75.9 | 78.6 | 28.7 | 70.0 |
| **Spindle** | **86.9** | **75.5** | **79.8** | **79.4** | **69.8** | **78.3** | **80.1** | **37.3** | **73.3** |
| Bottleneck | 84.5 | 74.6 | 76.9 | 78.1 | 69.2 | 76.2 | 78.6 | 32.1 | 71.3 |

Table 2: Average performance (low-compute, low-epoch regime: 100 random models, 3 tuning epochs) on the GLUE datasets using the T5-base pretrained backbone model. We compare adding different parameter allocation constraints to the $\mathcal{S}_1$ design space.

| Param Allocation | SST-2 | MNLI | QNLI | QQP | RTE | STS-B | MRPC | CoLA | Avg |
|---|---|---|---|---|---|---|---|---|---|
| Increasing | 87.2 | **77.9** | 79.4 | 78.7 | 71.6 | 77.6 | **81.4** | 32.0 | 73.2 |
| **Uniform** | **87.8** | 77.4 | **80.1** | **80.5** | **73.9** | **78.1** | 80.4 | 34.3 | **74.0** |
| Decreasing | 86.4 | 75.8 | 78.4 | 77.0 | 70.4 | 77.1 | 78.7 | **35.8** | 72.4 |

**Uniform** $(N_{i+1} = N_i)$: the number of layers in groups is the same;
**Decreasing** $(N_{i+1} < N_i)$: the number of layers in groups gradually decreases;
**Spindle** $(N_1 < N_2 = N_3 > N_4)$: the numbers of layers in groups at both ends are smaller;
**Bottleneck** $(N_1 > N_2 = N_3 < N_4)$: the numbers of layers in groups at both ends are bigger.

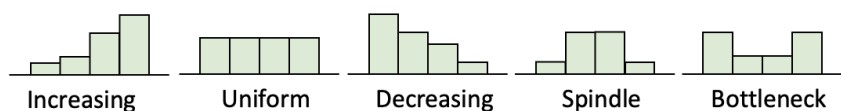

Figure 2: Layer grouping patterns: group ID $(G_1, \ldots G_4)$ vs. number of layers per group.

These layer grouping patterns lead to 5 possible design choices. They consist of all models in the $\mathcal{S}_0$ design space that satisfy one of these grouping pattern constraints. To compare the overall model qualities of different design spaces, we (i) randomly sample 100 models from the $\mathcal{S}_0$ design space that satisfy each grouping pattern constraint (Figure 2); (ii) fine-tune with 3 epochs; and (iii) compute the average performance for each design space. We will follow this procedure as we progressively add new constraints later.

The average performance is shown in Table 1 [4]. We find that models from the design space with the spindle grouping pattern (Figure 2) consistently outperform those from the other design spaces across all the 8 GLUE tasks. In other words, we find that fine-tuning works better if we treat a small number of layers close to the input and close to the output as special, and furthermore, if we divide up the bulk of the network into two blocks, each with their own design choices.

*Applying the spindle grouping partitioning to $\mathcal{S}_0$ yields the new design space $\mathcal{S}_1$.*

### 4.3 $\mathcal{S}_2$ — VARYING THE NUMBER OF TRAINABLE PARAMETERS PER LAYER

Now that we know how to group the layers we need to establish how to allocate the parameters $n_i$ within the layers $i$ of each group. In particular, we consider the following options:

**Increasing** $(n_{i+1} \geq n_i)$: number of trainable parameters per layer increases or remains the same.

---

[4]The training time for the step is shown in the Table 18 in the Appendix.

Table 3: Average performance (low-compute, low-epoch regime: 100 random models, 3 tuning epochs) on the GLUE datasets using the T5-base pretrained backbone model. We compare adding different tunable group constraints to the $\mathcal{S}_2$ design space.

| Tunable Groups | SST-2 | MNLI | QNLI | QQP | RTE | STS-B | MRPC | CoLA | Avg |
|---|---|---|---|---|---|---|---|---|---|
| $G_1$ | 82.6 | 72.1 | 77.6 | 70.6 | 65.3 | 71.9 | 77.6 | 27.6 | 68.2 |
| $G_2$ | 83.3 | 72.8 | 77.5 | 72.8 | 63.6 | 72.8 | 77.5 | 27.5 | 68.4 |
| $G_3$ | 83.6 | 73.3 | 78.2 | 73.3 | 66.4 | 71.3 | 77.9 | 22.9 | 68.4 |
| $G_4$ | 83.2 | 73.0 | 77.9 | 73.7 | 63.9 | 72.0 | 77.9 | 27.9 | 68.7 |
| $G_1, G_2$ | 83.5 | 73.2 | 78.0 | 75.4 | 67.7 | 73.2 | 78.0 | 28.0 | 69.6 |
| $G_3, G_4$ | 87.8 | 74.6 | 78.3 | 76.9 | 68.6 | 74.3 | 78.3 | 28.3 | 70.7 |
| $G_1, G_2, G_3$ | 86.0 | 75.8 | 79.0 | 77.8 | 71.8 | 78.8 | 79.0 | 33.0 | 72.6 |
| $G_2, G_3, G_4$ | 85.2 | 76.6 | 79.1 | 78.6 | 70.1 | 77.6 | 79.1 | 31.9 | 72.2 |
| $\boldsymbol{G_1, G_2, G_3, G_4}$ | **88.3** | **77.4** | **82.1** | **81.5** | **74.9** | **79.4** | **81.4** | **34.3** | **74.9** |

**Uniform** ($n_{i+1} = n_i$): number of trainable parameters in every layer is constant;
**Decreasing** ($n_{i+1} \leq n_i$): number of trainable parameters per layer decreases or remains the same.

As above, we obtain 100 models for each of these 3 new design spaces. Table 2 reports the average performance of these 3 design spaces. The uniform allocation design pattern obtains the highest GLUE average performance, making this relatively simple, interpretable design pattern favorable.

*Allocating the number of trainable parameters to layers uniformly yields the new design space $\mathcal{S}_2$.*

## 4.4   $\mathcal{S}_3$ — SELECTING THE GROUPS

Given that we established how to partition layers into groups, and how to allocate parameters per group, the next step is to assess whether all groups actually need tuning. Rather than exploring the $2^4 - 1 = 15$ combinatorial choices we limit ourselves to the $4(4 + 1)/2 = 10$ options with the exception of $(G_2, G_3)$, since focusing on interior groups only does not yield good results (this is consistent with our findings in Table 3).

Based on the GLUE average performance, we find that all the groups need to be tuned to obtain the best results. This suggests that all the groups of pretrained layers have captured useful information that should be adapted to the downstream tasks.

*Tuning all the groups yields the new design space $\mathcal{S}_3$.*

## 4.5   $\mathcal{S}_4$ — SELECTING STRATEGIES PER GROUP

So far the structure we've been exploring is fairly trivial: $\mathcal{S}_4$ amounts to a uniform distribution of parameters over the layers of the groups and to tuning all groups. This belies the fact that we still have significant freedom of design in picking specific fine-tuning approaches. Specifically, each design space consists of models that assign a subset of {Adapter (A), Prefix (P), BitFit (B), and LoRA (L)} to the layers of each group $G_i$ for $i \in \{1 \ldots 4\}$. This is quite a large space of options. To make some headway, we determine the ideal choice progressively by first reviewing strategies for $G_1$, then $G_2$ up to $G_4$. Due to space constraints the details of this procedure are relegated to the appendix($G_1$ in Table 8, $G_2$ Table 9, $G_3$ in Table 10, and $G_4$ in Table 11). We arrive at the following strategy assignment for the T5-base pretrained backbone:

$$G_1\text{: (A, L)} — G_2\text{: (A, P)} — G_3\text{: (A, P, B)} — G_4\text{:(P, B, L)}$$

For example, Adapter is more recommended in groups closer to input, while BitFit is more recommended in groups closer to the output. *The resulting design space will be referred to as $\mathcal{S}_4$.*

## 4.6   VERIFICATION OF THE DESIGN CHOICES ON T5-3B

So far our results have led to a competent fine-tuning strategy for T5-base. To assess whether we actually *discovered* some useful strategies that have validity beyond T5-base, we need to apply it to

other models, too. For convenience we pick T5-3b. As before, the detailed results are relegated to the appendix (Tables 12, 13, 14 and 15). We observe that the following design patterns still apply:

1. grouping layers in a spindle pattern (Table 12)
2. uniformly allocating the number of trainable parameters to layers (Table 13)
3. tuning all the groups (Table 14)
4. tuning different groups with proper strategies (Table 15)

Note that for T5-3b (with final design space $\mathcal{S}_4$-3b), the discovered proper strategy assignment is slightly different

$$G_1: \text{(P, L)} — G_2: \text{(A, L)} — G_3: \text{(P, B, L)} — G_4: \text{(A, P, B)}.$$

### 4.7 EXPERIMENTAL SETUP

**Datasets.** Our process is based on the average performance on the widely-used GLUE benchmark (Wang et al., 2018). It covers a wide range of natural language understanding tasks. First, *single-sentence tasks* include (i) Stanford Sentiment Treebank (SST-2) and (ii) Corpus of Linguistic Acceptability (CoLA). Second, *similarity and paraphrase tasks* include (i) Quora Question Pairs (QQP), (ii) Semantic Textual Similarity Benchmark (STS-B), and (iii) Microsoft Research Paraphrase Corpus (MRPC). Third, *inference tasks* include (i) Multi-Genre Natural Language Inference (MNLI), (ii) Question Natural Language Inference (QNLI), and (iii) Recognizing Textual Entailment (RTE). To compare performance, the Matthews correlation is measured for CoLA; the Spearman correlation is used for STS-B, and accuracy is measured for the remaining GLUE tasks.

**Pretrained Backbone Models and Model Settings** We use T5-base/3b (Raffel et al., 2020) as the main pretrained backbone models for discovering design patterns via our parameter-efficient fine-tuning design spaces. We use HuggingFace Transformers for our implementations and follow the default settings. During the exploration, we fix the total number of trainable parameters (in the percentage of that in the backbone model) by following He et al. (2022).

By limiting ourselves to a rather concise parameter space and a small number of parameters within that parameter space that we allow to be fine-tuned we ensure that exploration remains computationally feasible. Obviously, this exploration would be pointless *if* the discovered insights were not portable. Hence, we need to evaluate how well the strategies perform on new models and new architectures.

## 5 EVALUATION

The $\mathcal{S}_4$ model (Section 4.5) and $\mathcal{S}_4$-3b model (Section 4.6) adopt the design patterns discovered from T5-base and T5-3b, respectively. We will evaluate their effectiveness when applied to different pretrained backbones and different NLP tasks.

### 5.1 EXPERIMENTAL SETUP

**Dataset.** Besides the GLUE datasets (Wang et al., 2018) (Section 4.7), we evaluate our methods on two generation tasks used by He et al. (2022): *Abstractive Summarization* using XSum (Narayan et al., 2018), and *Machine Translation* using the WMT 2016 en-ro dataset (Bojar et al., 2016). We report ROUGE scores (Lin, 2004) on the XSum test set, and BLEU scores (Papineni et al., 2002) on the en-ro test set.

**Models and Model Settings.** We mainly compare our methods with the following baselines: (i) **Full Fine-tuning** (full): it fine-tunes all the model parameters in the pretrained models; (ii) **Adapter** (Houlsby et al., 2019b): it adds adapter modules to each transformer layer; (iii) **Prefix** (Li & Liang, 2021): it optimizes a set of small continuous vectors prepended to transformer layers; (iv) **BitFit** (Zaken et al., 2021): it only updates the bias terms in pretrained models; (v) **LoRA** (Hu et al., 2021): it decomposes the attention weight into low-rank matrices to reduce the number of trainable parameters. Besides T5 (Raffel et al., 2020), we additionally apply our methods to other backbone models including RoBERTa-base/large (Liu et al., 2019) and BART-base/large (Lewis et al., 2020a). We use the default settings. We set the total number of trainable parameters (in the percentage of

Table 4: Performances of different tuning methods on the GLUE datasets using the T5-base (upper part) and T5-3b (lower part) pretrained backbone models, respectively. The results are averaged over 20 random runs (with standard deviations as subscripts). The $\mathcal{S}_4$-model and the $\mathcal{S}_4$-3b-model perform significantly better than the second-best PEFT methods in all the eight datasets at the significance level $p < 0.05(*)$ or even $p < 0.01(**)$.

| Method | SST-2 | MNLI | QNLI | QQP | RTE | STS-B | MRPC | CoLA | Average |
|---|---|---|---|---|---|---|---|---|---|
| full | 95.2 | 87.1 | 93.7 | 89.4 | 80.1 | 89.4 | 90.7 | 51.1 | 84.5 |
| Adapter | 94.6 | 85.5 | 89.8 | 86.7 | 75.3 | 86.7 | 89.1 | 59.2 | 83.3 |
| Prefix | 94.0 | 81.6 | 87.8 | 83.4 | 64.3 | 83.1 | 84.8 | 34.0 | 76.6 |
| BitFit | 94.4 | 84.5 | 90.6 | 88.3 | 74.3 | 86.6 | 90.1 | 57.7 | 83.3 |
| LoRA | 94.8 | 84.7 | 91.6 | 88.5 | 75.8 | 86.3 | 88.7 | 51.5 | 82.7 |
| $\mathcal{S}_4$-model | $\mathbf{95.5}^{**}_{1.7}$ | $\mathbf{87.6}^{**}_{1.0}$ | $\mathbf{92.7}^{**}_{1.1}$ | $\mathbf{88.8}^{**}_{1.0}$ | $\mathbf{80.4}^{*}_{2.3}$ | $\mathbf{87.4}^{*}_{2.0}$ | $\mathbf{91.2}^{**}_{2.4}$ | $\mathbf{62.2}^{*}_{3.2}$ | **85.7** |
| full | 97.4 | 91.4 | 96.3 | 89.7 | 91.1 | 90.6 | 92.5 | 67.1 | 89.5 |
| Adapter | 96.3 | 89.9 | 94.7 | 87.8 | 83.4 | 90 | 89.7 | 65.2 | 87.1 |
| Prefix | 96.3 | 82.8 | 88.9 | 85.5 | 78.3 | 83.5 | 85.4 | 42.7 | 80.4 |
| BitFit | 95.8 | 89.5 | 93.5 | 88.5 | 86.2 | 90.7 | 88.6 | 64.2 | 87.1 |
| LoRA | 96.2 | 90.6 | 94.9 | 89.1 | 91.2 | 91.1 | 91.1 | 67.4 | 88.9 |
| $\mathcal{S}_4$-3b-model | $\mathbf{97.2}^{**}_{1.8}$ | $\mathbf{91.6}^{**}_{1.2}$ | $\mathbf{96.6}^{**}_{1.0}$ | $\mathbf{89.5}^{**}_{1.5}$ | $\mathbf{91.5}^{*}_{2.8}$ | $\mathbf{91.5}^{*}_{2.5}$ | $\mathbf{91.9}^{*}_{2.0}$ | $\mathbf{69.7}^{*}_{3.4}$ | **89.9** |

Table 5: Performances of different tuning methods on GLUE datasets using the RoBERTa-base (upper part) and RoBERTa-large (lower part) pretrained backbone models. The results are averaged over 20 random runs (with standard deviations as subscripts). Here we also include two baselines: (i) $\mathcal{S}_0$-*model*, where all the designs are randomly selected for RoBERTa as in the $\mathcal{S}_0$ design space; (ii) $\mathcal{S}_3$-*model*, where strategies are randomly assigned to different RoBERTa layer groups as in the $\mathcal{S}_3$ design space. The $\mathcal{S}_4$-model and $\mathcal{S}_4$-3b-model perform significantly better than the second-best PEFT methods in all the eight datasets at the significance level $p < 0.05(*)$ or even $p < 0.01(**)$.

| Method | SST-2 | MNLI | QNLI | QQP | RTE | STS-B | MRPC | CoLA | Average |
|---|---|---|---|---|---|---|---|---|---|
| full | 94.8 | 87.6 | 92.8 | 91.9 | 80.8 | 90.3 | 90.2 | 63.6 | 86.5 |
| Adapter | 94.2 | 87.1 | 93.1 | 90.2 | 71.5 | 89.7 | 88.5 | 60.8 | 84.4 |
| Prefix | 94.0 | 86.8 | 91.3 | 90.5 | 74.5 | 90.3 | 88.2 | 61.5 | 84.6 |
| BitFit | 93.7 | 84.8 | 91.3 | 84.5 | 77.8 | **90.8** | 90.0 | 61.8 | 84.3 |
| LoRA | **94.9** | 87.5 | 93.1 | 90.8 | 83.1 | 90.0 | 89.6 | 62.6 | 86.4 |
| $\mathcal{S}_0$-model | 94.2 | 95.3 | 90.4 | 90.6 | 75.6 | 89.6 | 88.0 | 60.9 | 85.6 |
| $\mathcal{S}_3$-model | 94.3 | 87.2 | 92.8 | 91.0 | 81.8 | 90.3 | 89.2 | 63.2 | 86.2 |
| $\mathcal{S}_4$-model | $94.8_{1.6}$ | $\mathbf{87.8}^{**}_{0.8}$ | $\mathbf{93.4}^{**}_{1.3}$ | $\mathbf{91.6}^{*}_{1.2}$ | $\mathbf{85.8}^{**}_{1.8}$ | $\mathbf{90.4}^{*}_{2.0}$ | $\mathbf{90.0}^{**}_{1.8}$ | $\mathbf{63.2}^{*}_{3.5}$ | **87.1** |
| full | 96.4 | 90.2 | 94.7 | 92.2 | 86.6 | 92.4 | 90.9 | 68.0 | 88.9 |
| Adapter | 96.6 | 90.5 | 94.8 | 91.7 | 80.1 | 92.1 | 90.9 | 67.8 | 88.1 |
| Prefix | 95.7 | 87.6 | 92.1 | 88.7 | 82.3 | 89.6 | 87.4 | 62.8 | 85.7 |
| BitFit | 96.1 | 88.0 | 93.4 | 90.2 | 86.2 | 90.9 | **92.7** | 64.2 | 87.7 |
| LoRA | 96.2 | 90.6 | 94.7 | 91.6 | **87.4** | 92.0 | 89.7 | 68.2 | 88.8 |
| $\mathcal{S}_0$-model | 95.5 | 86.5 | 92.3 | 89.8 | 84.6 | 89.2 | 86.3 | 61.2 | 85.6 |
| $\mathcal{S}_3$-model | 96.3 | 89.4 | 93.8 | 90.2 | 85.9 | 90.8 | 90.9 | 63.4 | 87.6 |
| $\mathcal{S}_4$-3b-model | $\mathbf{96.6}^{**}_{1.3}$ | $\mathbf{90.8}^{*}_{1.1}$ | $\mathbf{95.1}^{**}_{0.8}$ | $\mathbf{92.0}^{*}_{1.2}$ | $87.2_{2.8}$ | $\mathbf{92.3}^{*}_{2.2}$ | $\mathbf{91.8}^{**}_{1.8}$ | $\mathbf{68.4}^{*}_{3.2}$ | **89.3** |

that in the backbone model) by following He et al. (2022). Specifically, this value is set to 0.5% for Adapter, Prefix, LoRA, and our methods, and 0.1% for BitFit.

For all the experiments, we followed Liu et al. (2019) to set the linear decay scheduler with a warmup ratio of 0.06 for training. The batch size was 128 for base models and 64 for large models. The maximum learning rate was $5e - 5$ and the maximum number of training epochs was set to be either 5 or 10. All the experiments were performed using 8 A100 GPUs.

## 5.2 EFFECTIVENESS ON DIFFERENT BACKBONES

**GLUE with T5 Backbone.** With our discovered design patterns, we fine-tune T5-base ($\mathcal{S}_4$-model) and T5-3b ($\mathcal{S}_4$-3b-model) on GLUE and compare them with all the baseline methods. The results are shown in Table 4, where the key measure is the GLUE average performance (last column). We

Table 6: Performance of different tuning methods on generation tasks (XSUM and en-ro) using the BART-base (left) and BART-large (right) pretrained backbone models.

| BART-base | XSUM(R-1/2/L) | en-ro (BLEU) | BART-large | XSUM(R-1/2/L) | en-ro (BLEU) |
|---|---|---|---|---|---|
| full | 40.5/19.2/34.8 | 34.5 | full | 45.1/22.3/37.2 | 37.9 |
| Adapter | 37.7/17.9/33.1 | 33.3 | Adapter | 43.8/20.8/35.7 | 35.3 |
| Prefix | 38.2/18.4/32.4 | 33.8 | Prefix | 43.4/20.4/35.5 | 35.6 |
| BitFit | 37.2/17.5/31.4 | 33.2 | BitFit | 42.8/18.7/33.2 | 35.2 |
| LoRA | 38.9/18.6/33.5 | 33.6 | LoRA | 42.9/19.4/34.8 | 35.8 |
| PA | 39.3/18.7/33.8 | 33.8 | PA | 43.9/20.6/35.6 | 36.4 |
| $\mathcal{S}_4$-**model** | **40.2/19.3/34.2** | **34.1** | $\mathcal{S}_4$-**3b-model** | **44.3/21.7/36.8** | **37.2** |

find that our $\mathcal{S}_4$-model and $\mathcal{S}_4$-3b-model consistently outperform the investigated methods in the key measure. By tuning only $0.5\%$ parameters, our methods even outperform the full fine-tuning baseline where all the parameters are tuned, indicating the effectiveness of our discovered parameter-efficient fine-tuning design patterns.

**GLUE with RoBERTa Backbone.** We directly apply the $\mathcal{S}_4$-model and $\mathcal{S}_4$-3b-model (adopting design patterns discovered using T5-base and T5-3b) to fine-tune the RoBERTa-base and RoBERTa-large pretrained backbone models, respectively. We keep all the other settings the same and evaluate them on GLUE datasets. We also compare with variant methods randomly sampled from two design spaces: (i) $\mathcal{S}_0$-*model*, where all the designs are randomly selected for RoBERTa as in $\mathcal{S}_0$; (ii) $\mathcal{S}_3$-*model*, where strategies are randomly assigned to different RoBERTa layer groups as in $\mathcal{S}_3$. Table 5 shows that (i) the design patterns (adopted by $\mathcal{S}_4$-model and $\mathcal{S}_4$-3b-model) discovered using T5 models are applicable to the RoBERTa backbone models and outperform the investigated methods in GLUE average performance with no extra discovery process[5]; (ii) improved performance from $\mathcal{S}_0$-models, $\mathcal{S}_3$-models, to $\mathcal{S}_4$-(3b)-models support adding more constraints in the pattern discovery process (Section 4).

**SuperGLUE with XLNet Backbone.** We also directly use the $\mathcal{S}_4$-model and $\mathcal{S}_4$-3b-model (adopting design patterns discovered using T5-base and T5-3b) to fine-tune the XLNet-base and XLNet-large pretrained backbone models without any extra discovery process. We keep all the other settings the same and evaluate them on SuperGLUE datasets. Table 17 (In the Appendix) reiterates the fact that our PEFT design patterns discovered from T5 models are generelizable to the XLNet backbone models and outperform the investigated methods in other tasks (SuperGLUE) with no additional discovery process.

**Generation Tasks with BART Backbone.** We further apply the $\mathcal{S}_4$-model and $\mathcal{S}_4$-3b-model (adopting design patterns discovered using T5-base and T5-3b) to fine-tune the BART-base and BART-large pretrained backbone models, respectively. We evaluate the models on two generation tasks: summarization (XSUM) and machine translation (en-ro) following He et al. (2022). We also compare with PA (parallel adapter) using the same number of trainable parameters (He et al., 2022). Table 6 shows that our methods, although adopting design patterns discovered from classification tasks using T5, still outperform investigated parameter-efficient fine-tuning strategies on generation tasks with different BART backbones.

## 6 CONCLUSION

Parameter-efficient fine-tuning adapts knowledge in pretrained models to down-stream tasks in a more parameter-efficient fashion. Instead of focusing on designing another strategy in the first place, we introduced parameter-efficient fine-tuning design spaces. We empirically discovered several design patterns in parameter-efficient fine-tuning. These design patterns led to new parameter-efficient fine-tuning methods. Experiments showed that these methods consistently outperform investigated parameter-efficient fine-tuning strategies across different backbone models and different tasks in natural language processing.

---

[5]Future works might repeat the discovery process using RoBERTa to improve performance for this backbone.

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

# A  MORE EXPERIMENTAL RESULTS

Table 7: Average performances (low-compute, low-epoch regime: 100 random models, tuning epochs = 1, 2, 3, 4, 20 for five different blocks) on the GLUE datasets using the T5-base pretrained backbone model. We compare adding different grouping constraints to the $\mathcal{S}_0$ design space.

| Grouping Patterns | SST-2 | MNLI | QNLI | QQP | RTE | STS-B | MRPC | CoLA | Avg |
|---|---|---|---|---|---|---|---|---|---|
| | | | | 1 epochs | | | | | |
| **Increasing** | 73.2 | 63.3 | **67.8** | 68.8 | **63.8** | 67.2 | 64.1 | 11.0 | **59.9** |
| Uniform | **72.8** | 64.1 | 63.4 | 63.4 | 62.5 | **69.8** | 65.8 | 12.1 | 59.2 |
| Decreasing | 72.4 | 63.2 | 65.1 | 69.8 | 59.3 | 62.7 | 63.6 | **18.7** | 59.4 |
| Spindle | 72.6 | **64.8** | 66.8 | **71.1** | 62.1 | 62.3 | 64.8 | 12.3 | 59.6 |
| Bottleneck | 72.2 | 63.7 | 65.3 | 68.3 | 61.2 | 63.2 | **66.6** | 12.1 | 59.0 |
| | | | | 2 epochs | | | | | |
| Increasing | 76.2 | 69.3 | 73.2 | 76.5 | 65.8 | 72.2 | **74.0** | 21.0 | 66.0 |
| Uniform | 74.8 | 70.9 | **74.1** | 75.6 | 66.5 | 73.4 | 71.2 | 22.1 | 66.1 |
| Decreasing | 71.4 | 70.1 | 72.1 | **76.8** | 64.3 | 71.7 | 73.6 | 18.7 | 64.8 |
| **Spindle** | **76.6** | **71.9** | 71.8 | 74.4 | **67.5** | **73.5** | 71.8 | 22.3 | **66.2** |
| Bottleneck | 74.2 | 71.1 | 69.6 | 73.3 | 65.2 | 73.3 | 73.6 | **24.1** | 65.5 |
| | | | | 3 epochs | | | | | |
| Increasing | 85.3 | 74.9 | 77.2 | 77.5 | 66.8 | 76.2 | 76.0 | 33.0 | 70.8 |
| Uniform | 84.8 | 73.7 | 78.1 | 78.6 | 68.5 | 77.8 | 79.2 | 36.1 | 72.1 |
| Decreasing | 81.9 | 72.1 | 78.3 | 76.7 | 67.3 | 75.9 | 78.6 | 28.7 | 69.9 |
| **Spindle** | **86.9** | **75.5** | **79.8** | **79.4** | **69.8** | **78.3** | **80.1** | **47.3** | **74.6** |
| Bottleneck | 84.5 | 74.6 | 76.9 | 78.1 | 69.2 | 76.2 | 78.6 | 32.1 | 71.3 |
| | | | | 4 epochs | | | | | |
| Increasing | 88.3 | 78.5 | 80.2 | 80.5 | 70.8 | 80.2 | 80.0 | 37.0 | 74.4 |
| Uniform | 88.8 | 78.9 | 81.9 | 81.5 | 71.5 | 80.8 | 81.4 | 39.1 | 75.4 |
| Decreasing | 87.6 | 74.1 | 80.8 | 81.7 | 79.3 | 78.9 | 79.6 | 38.7 | 75.1 |
| **Spindle** | **89.6** | **79.8** | **83.6** | **82.8** | **71.8** | **81.3** | **82.1** | **39.3** | **76.3** |
| Bottleneck | 86.5 | 77.6 | 82.7 | 81.1 | 70.2 | 70.9 | 81.6 | 36.1 | 73.3 |
| | | | | 20 epochs | | | | | |
| Increasing | 92.3 | 83.3 | 86.2 | 82.5 | 71.8 | 82.2 | 84.0 | 51.0 | 79.1 |
| Uniform | 92.8 | 83.9 | 86.1 | 83.6 | 72.5 | 83.8 | 84.2 | 52.1 | 79.9 |
| Decreasing | 91.4 | 82.1 | 85.1 | 83.1 | 69.3 | 81.7 | 83.6 | 48.7 | 78.1 |
| **Spindle** | **93.6** | **84.8** | **87.8** | **84.4** | **73.5** | **84.3** | **85.8** | **52.3** | **80.8** |
| Bottleneck | 92.1 | 82.6 | 85.6 | 83.3 | 71.2 | 83.2 | 84.6 | 52.1 | 79.3 |

# B  GENERAL EFFECTIVENESS ON SUPERGLUE WITH XLNET BACKBONES

We also directly use the $\mathcal{S}_4$-model and $\mathcal{S}_4$-3b-model (adopting design patterns discovered using T5-base and T5-3b) to fine-tune the XLNet-base and XLNet-large pretrained backbone models without any extra discovery process. We keep all the other settings the same and evaluate them on SuperGLUE datasets. Table 17 reiterates the fact that our PEFT design patterns discovered from T5 models are generelizable to the XLNet backbone models and outperform the investigated methods in other tasks (SuperGLUE) with no additional discovery process.

Table 8: Average performances (low-compute, low-epoch regime: 100 random models, 3 tuning epochs) on the GLUE datasets using the T5-base pretrained backbone model. We compare adding different $G_1$ strategy assignment constraints to the $\mathcal{S}_3$ design space.

| Strategy Assignment | SST-2 | MNLI | QNLI | QQP | RTE | STS-B | MRPC | CoLA | Avg |
|---|---|---|---|---|---|---|---|---|---|
| $G_1$-Adapter (A) | 89.8 | 83.5 | 84.9 | 80.8 | 72.5 | 80.8 | **78.5** | **37.7** | 76.1 |
| $G_1$-Prefix (P) | 89.3 | 83.1 | 84.4 | 80.1 | 70.1 | 80.0 | 77.6 | 33.0 | 74.7 |
| $G_1$-BitFit (B) | 89.0 | 82.9 | 84.1 | 81.4 | 72.0 | 81.1 | 77.0 | 30.8 | 74.8 |
| $G_1$-LoRA (L) | 89.9 | 83.6 | 85.0 | 81.1 | 71.8 | 81.0 | 78.8 | 35.3 | 75.8 |
| $G_1$-(P, L) | 89.1 | 82.8 | 85.1 | 81.2 | 71.9 | 81.5 | 79.1 | 35.0 | 75.7 |
| $G_1$-(A, P) | 89.8 | 82.8 | 84.8 | 81.1 | 72.2 | 81.3 | 79.2 | 36.4 | 75.9 |
| $G_1$-**(A, L)** | 89.6 | **83.8** | **85.6** | 81.3 | **72.9** | **81.7** | **79.5** | 36.8 | **76.4** |
| $G_1$-(A, P, L) | 89.6 | 83.5 | 85.2 | 81.5 | 72.2 | 81.4 | 79.2 | 35.2 | 75.9 |
| $G_1$-(P, B, L) | 89.3 | 83.6 | 85.5 | 81.6 | 72.3 | 81.0 | 78.8 | 35.7 | 76.0 |
| $G_1$-(A, P, B) | 89.2 | 83.3 | 84.8 | **81.8** | 72.5 | 81.1 | 78.6 | 35.6 | 75.8 |
| $G_1$-(A, B, L) | 89.8 | 83.4 | 84.8 | 81.1 | 72.6 | 81.6 | 79.4 | 34.8 | 75.9 |
| $G_1$-(A, P, B, L) | **90.0** | 83.1 | 85.3 | 81.6 | 72.6 | 81.4 | 79.2 | 36.5 | 76.1 |

Table 9: Average performances (low-compute, low-epoch regime: 100 random models, 3 tuning epochs) on the GLUE datasets using the T5-base pretrained backbone model. We compare adding different $G_2$ strategy assignment constraints with $G_1$-(L, A) to the $\mathcal{S}_3$ design space.

| Strategy Assignment | SST-2 | MNLI | QNLI | QQP | RTE | STS-B | MRPC | CoLA | Avg |
|---|---|---|---|---|---|---|---|---|---|
| $G_2$-Adapter (A) | 91.6 | 84.3 | 85.5 | **82.3** | 73.5 | 82.8 | 81.3 | 38.8 | 77.5 |
| $G_2$-Prefix (P) | 89.6 | 84.0 | 86.5 | 81.5 | 73.3 | 82.5 | 80.5 | 36.2 | 76.7 |
| $G_2$-BitFit (B) | 91.2 | 83.6 | 85.7 | 82.9 | 72.6 | 82.6 | 80.8 | 33.1 | 76.5 |
| $G_2$-LoRA (L) | 91.4 | 84.4 | 86.1 | 82.0 | 72.8 | 81.8 | 81.6 | 39.8 | 77.4 |
| $G_2$-(P, L) | 91.6 | 84.6 | 86.8 | 81.8 | 73.8 | 82.8 | 82.0 | 38.5 | 77.7 |
| $G_2$-**(A, P)** | **92.2** | **84.2** | **87.1** | 82.2 | **74.4** | 83.0 | **82.5** | 40.8 | **78.3** |
| $G_2$-(A, L) | 92.0 | 84.4 | 86.5 | 81.8 | 73.6 | 82.6 | 82.2 | 40.1 | 77.9 |
| $G_2$-(A, P, L) | 91.8 | 84.8 | 86.8 | 81.8 | 74.1 | 83.0 | 82.1 | 37.9 | 77.7 |
| $G_2$-(P, B, L) | 91.6 | 84.1 | 87.1 | 82.0 | 74.0 | 82.9 | 82.4 | 35.8 | 77.4 |
| $G_2$-(A, P, B) | 91.8 | 84.2 | 86.8 | 82.1 | 73.7 | **83.3** | 82.2 | 41.2 | 78.1 |
| $G_2$-(A, B, L) | **92.2** | 84.3 | 86.1 | 82.0 | 74.1 | 83.2 | 82.0 | 37.6 | 77.6 |
| $G_2$-(A, P, B, L) | 92.0 | 84.1 | 87.0 | 81.9 | 74.2 | 83.1 | 81.3 | **42.4** | 78.1 |

Table 10: Average performances (low-compute, low-epoch regime: 100 random models, 3 tuning epochs) on the GLUE datasets using the T5-base pretrained backbone model. We compare adding different $G_3$ strategy assignment constraints with $G_1$-(L, A) – $G_2$-(P, A) to the $\mathcal{S}_3$ design space.

| Strategy Assignment | SST-2 | MNLI | QNLI | QQP | RTE | STS-B | MRPC | CoLA | Avg |
|---|---|---|---|---|---|---|---|---|---|
| $G_3$-Adapter (A) | 92.5 | 85.3 | 87.5 | **83.3** | 73.9 | 84.0 | 83.8 | **44.9** | 79.4 |
| $G_3$-Prefix (P) | 91.5 | 84.7 | 86.7 | 82.6 | 74.2 | 83.8 | 82.9 | 40.5 | 78.4 |
| $G_3$-BitFit (B) | 91.9 | 84.3 | 87.0 | 82.0 | 73.6 | 84.1 | 83.3 | 36.1 | 77.8 |
| $G_3$-LoRA (L) | 92.8 | 85.4 | 87.8 | 83.5 | 74.7 | 82.4 | 84.0 | 44.0 | 79.3 |
| $G_3$-(P, L) | 93.0 | 85.2 | 88.3 | 83.8 | 75.2 | 84.4 | 84.2 | 37.9 | 79.0 |
| $G_3$-(A, P) | 92.4 | 85.6 | 88.1 | 83.6 | 75.0 | 84.2 | 84.0 | 41.8 | 79.3 |
| $G_3$-(A, L) | 92.0 | 85.9 | 88.2 | 83.1 | 75.3 | 84.3 | 83.9 | 42.2 | 79.4 |
| $G_3$-(A, P, L) | 92.6 | 86.0 | 87.5 | 83.4 | 75.6 | 84.6 | 83.5 | 43.9 | 79.6 |
| $G_3$-(P, B, L) | 92.7 | 85.8 | 87.2 | 83.7 | 75.2 | 84.5 | 83.8 | 40.8 | 79.2 |
| $G_3$-**(A, P, B)** | 93.3 | **85.8** | **88.6** | **84.0** | 75.5 | **84.9** | 84.1 | 42.1 | **79.8** |
| $G_3$-(A, B, L) | **93.7** | 86.5 | 88.0 | 83.2 | **75.8** | 84.2 | 84.2 | 39.7 | 79.4 |
| $G_3$-(A, P, B, L) | 93.3 | 85.6 | 87.7 | 83.8 | 75.2 | 84.3 | **84.4** | 41.6 | 79.4 |

Table 11: Average performances (low-compute, low-epoch regime: 100 random models, 3 tuning epochs) on the GLUE datasets using the T5-base pretrained backbone model. We compare adding different $G_4$ strategy assignment constraints with $G_1$-(A, L) − $G_2$-(A, P) − $G_3$-(A, P, B) to the $\mathcal{S}_3$ design space.

| Strategy Assignment | SST-2 | MNLI | QNLI | QQP | RTE | STS-B | MRPC | CoLA | Avg |
|---|---|---|---|---|---|---|---|---|---|
| $G_4$-Adapter (A) | 93.8 | 85.8 | 88.6 | 84.8 | 76.3 | 85.8 | 86.0 | **48.5** | 81.2 |
| $G_4$-Prefix (P) | 93.5 | 85.2 | 88.3 | 83.6 | 76.8 | 85.3 | 85.6 | 44.8 | 80.3 |
| $G_4$-BitFit (B) | 94.1 | 85.3 | 88.9 | 84.4 | 77.1 | 85.4 | 86.2 | 46.1 | 80.9 |
| $G_4$-LoRA (L) | 94.0 | 86.0 | 89.2 | 85.0 | 77.2 | 85.5 | 85.8 | 47.7 | 81.3 |
| $G_4$-(P, L) | 94.3 | 86.2 | 89.3 | 85.8 | 78.0 | 86.0 | 88.2 | 47.2 | 81.8 |
| $G_4$-(A, P) | 94.1 | 86.2 | 89.6 | 85.4 | 77.9 | 86.2 | 86.9 | 45.3 | 81.4 |
| $G_4$-(A, L) | 94.2 | 85.9 | 89.2 | 85.5 | 77.8 | 86.2 | 88.0 | 46.8 | 81.7 |
| $G_4$-(A, P, L) | 94.1 | 85.8 | 88.8 | 85.7 | 77.4 | 86.5 | 87.9 | 44.8 | 81.3 |
| $G_4$-**(P, B, L)** | **94.6** | **86.4** | **90.4** | **86.1** | 78.2 | **86.8** | **88.5** | 47.2 | **82.3** |
| $G_4$-(A, P, B) | 94.5 | 86.0 | 89.6 | 86.0 | 78.0 | 86.2 | 88.1 | 44.8 | 81.6 |
| $G_4$-(A, B, L) | 94.3 | **86.4** | 89.2 | 85.6 | 78.2 | 86.4 | 88.3 | 46.6 | 81.9 |
| $G_4$-(A, P, B, L) | 94.2 | 86.2 | 89.2 | 85.9 | **78.5** | 86.1 | 88.0 | 45.3 | 81.6 |

Table 12: Average performances (low-compute, low-epoch regime: 100 random models, 3 tuning epochs) on the GLUE datasets using the T5-3b pretrained backbone model. We compare adding different layer grouping constraints to the $\mathcal{S}_0$ design space.

| Grouping Patterns | SST-2 | MNLI | QNLI | QQP | RTE | STS-B | MRPC | CoLA | Avg |
|---|---|---|---|---|---|---|---|---|---|
| $\mathcal{S}_0$-models | 80.3 | 72.1 | 74.7 | 72.8 | 76.9 | 75.2 | 71.0 | 32.2 | 69.4 |
| Increasing | 84.4 | 75.7 | 83.0 | 78.3 | 82.7 | 80.3 | 76.3 | 42.1 | 75.3 |
| Uniform | 86.8 | 77.1 | 82.6 | 76.2 | 83.8 | **81.6** | 77.3 | **48.9** | 76.8 |
| Decreasing | 83.2 | 74.3 | 81.8 | 77.3 | 82.8 | 79.9 | 76.5 | 40.8 | 74.5 |
| **Spindle** | **88.6** | **78.8** | **83.7** | 77.7 | **84.2** | 80.9 | **78.3** | 44.6 | **77.1** |
| Bottleneck | 86.3 | 77.0 | 82.2 | 75.6 | 83.3 | 80.2 | 77.1 | 41.5 | 75.4 |

Table 13: Average performances (low-compute, low-epoch regime: 100 random models, 3 tuning epochs) on the GLUE datasets using the T5-3b pretrained backbone model. We compare adding different layer parameter constraints to the $\mathcal{S}_1$ design space.

| Parameter Allocation | SST-2 | MNLI | QNLI | QQP | RTE | STS-B | MRPC | CoLA | Avg |
|---|---|---|---|---|---|---|---|---|---|
| Increasing | 90.3 | 79.3 | **84.9** | 79.3 | 85.2 | **82.8** | **79.2** | 50.1 | 78.9 |
| **Uniform** | **90.6** | **80.8** | 84.6 | **79.7** | **85.5** | 82.4 | 78.9 | **50.8** | **79.1** |
| Decreasing | 88.6 | 78.2 | 83.5 | 78.1 | 84.4 | 81.5 | 78.1 | 49.6 | 77.7 |

Table 14: Average performances (low-compute, low-epoch regime: 100 random models, 3 tuning epochs) on the GLUE datasets using the T5-3b pretrained backbone model. We compare adding different tuning groups constraints to the $\mathcal{S}_2$ design space.

| Tunable Groups | SST-2 | MNLI | QNLI | QQP | RTE | STS-B | MRPC | CoLA | Avg |
|:---:|:---:|:---:|:---:|:---:|:---:|:---:|:---:|:---:|:---:|
| $G_1$ | 88.3 | 78.3 | 82.2 | 77.4 | 82.1 | 80.7 | 76.1 | 49.4 | 76.8 |
| $G_2$ | 89.1 | 78.8 | 82.1 | 77.2 | 82.3 | 81.2 | 76.4 | 49.6 | 77.1 |
| $G_3$ | 89.6 | 78.5 | 82.6 | 78.1 | 83.8 | 81.9 | 77.4 | 48.7 | 77.5 |
| $G_4$ | 89.8 | 79.3 | 82.7 | 77.9 | 83.5 | 81.9 | 77.9 | 48.5 | 77.1 |
| $G_1, G_2$ | 90.1 | 80.2 | 83.4 | 78.5 | 84.3 | 82.4 | 78.5 | 51.1 | 78.5 |
| $G_3, G_4$ | 90.5 | 80.6 | 83.8 | 78.7 | 84.2 | 83 | 78.2 | 50.3 | 78.6 |
| $G_1, G_2, G_3$ | 90.6 | 80.3 | 84.9 | 79.3 | 84.7 | 82.9 | 79.3 | 50.2 | 79.0 |
| $G_2, G_3, G_4$ | 90.8 | 80.9 | 84.6 | 79.1 | 85.1 | 83.1 | 79.1 | 49.2 | 78.9 |
| $\boldsymbol{G_1, G_2, G_3, G_4}$ | **91.1** | **81.4** | **85.2** | **80.4** | **85.9** | **83.5** | **80.0** | **51.6** | **79.9** |

Table 15: Average performances (low-compute, low-epoch regime: 100 random models, 3 tuning epochs) on the GLUE datasets using the T5-3b pretrained backbone model. We compare adding different strategy assignment constraints following the process in Section 4.5.

| Strategy Assignment | SST-2 | MNLI | QNLI | QQP | RTE | STS-B | MRPC | CoLA | Avg |
|---|---|---|---|---|---|---|---|---|---|
| $G_1$-Adapter (A) | 91.1 | 81.4 | 86.1 | 80.5 | 86.7 | 83.3 | 80.1 | 50.8 | 80.0 |
| $G_1$-Prefix (P) | 90.8 | 81.1 | 85.5 | 80.2 | 86.2 | 83.1 | 79.8 | 50.2 | 79.6 |
| $G_1$-BitFit (B) | 90.2 | 81.3 | 85.1 | 79.6 | 85.8 | 82.8 | 79.6 | 49.5 | 79.2 |
| $G_1$-LoRA (L) | 91.4 | 81.9 | 86.2 | 80.8 | 86.4 | 83.9 | 80.8 | 49.6 | 80.0 |
| $G_1$-**(P, L)** | **91.8** | **82.9** | **86.8** | 81.3 | **87.1** | 84.2 | **81.6** | 52.3 | **81.0** |
| $G_1$-(A, P) | 91.3 | 81.9 | 86.4 | 81.1 | 85.6 | 83.7 | 80.7 | **52.8** | 80.1 |
| $G_1$-(A, L) | 91.6 | 82.3 | 86.1 | **81.5** | 85.8 | **84.9** | 81.5 | 51.8 | 80.6 |
| $G_1$-(A, P, L) | 91.1 | 81.7 | 85.8 | 81.2 | 86.4 | 84.2 | 80.9 | 52.3 | 80.4 |
| $G_1$-(P, B, L) | 91.5 | 82.8 | 86.3 | 81.4 | 86.1 | 83.6 | 81.2 | 51.5 | 80.5 |
| $G_1$-(A, P, B) | 91.3 | 82.3 | 86.7 | 80.8 | 86.8 | 84.3 | 80.7 | 51.8 | 80.5 |
| $G_1$-(A, B, L) | 91.7 | 82.5 | 86.2 | 81.3 | 86.3 | 84.6 | 81.3 | 51.7 | 80.7 |
| $G_1$-(A, P, B, L) | 91.6 | 82.3 | 86.2 | 81.1 | 86.6 | 84.2 | 81.1 | 51.1 | 80.5 |
| $G_2$-Adapter (A) | 92.1 | 82.5 | 86.4 | 81.8 | 87.2 | 84.8 | 81.8 | 53.8 | 81.3 |
| $G_2$-Prefix (P) | 91.8 | 83.1 | 87.2 | 81.6 | 86.2 | 84.4 | 81.1 | 52.8 | 81.0 |
| $G_2$-BitFit (B) | 91.2 | 82.1 | 86.4 | 81.1 | 86.3 | 84.6 | 80.3 | 53.1 | 80.6 |
| $G_2$-LoRA (L) | 92.6 | 82.9 | 87.5 | 81.3 | 87.4 | 85.1 | 81.9 | 52.2 | 81.4 |
| $G_2$-(P, L) | 91.6 | 82.7 | 87.6 | 81.6 | **87.8** | 85.3 | 82.1 | 52.8 | 81.4 |
| $G_2$-(A, P) | 92.1 | 83.3 | 87.5 | 81.9 | 87.4 | 85.5 | 81.8 | 53.1 | 81.5 |
| $G_2$-**(A, L)** | 92.5 | **83.7** | **88.1** | **82.2** | 87.4 | **85.7** | **82.9** | 53.6 | **82.1** |
| $G_2$-(A, P, L) | 92.3 | 83.4 | 87.4 | 81.6 | 87.1 | 85.3 | 81.4 | 53.2 | 81.4 |
| $G_2$-(P, B, L) | 91.8 | 83.1 | 87.4 | 81.5 | 87.2 | 85.1 | 82.7 | 53.8 | 81.5 |
| $G_2$-(A, P, B) | 91.5 | 82.6 | 87.8 | 81.3 | 86.5 | 85.2 | 82.1 | **54.2** | 81.4 |
| $G_2$-(A, B, L) | 92.6 | 83.5 | 87.2 | 82 | 87.3 | 86.5 | 82.5 | 52.8 | 81.8 |
| $G_2$-(A, P, B, L) | **92.8** | 83.2 | 87.6 | 81.6 | 87.5 | 85.5 | 82.4 | 51.2 | 81.5 |
| $G_3$-Adapter (A) | 92.6 | 84.1 | 88.3 | 81.8 | 87.8 | 85.4 | 82.8 | 55.2 | 82.2 |
| $G_3$-Prefix (P) | 92.1 | 83.3 | 87.6 | 81.4 | 87.1 | 85.4 | 82.6 | 53.5 | 81.6 |
| $G_3$-BitFit (B) | 92.4 | 83.9 | 88.4 | 82.1 | 87.2 | 85.8 | 82.4 | 53.3 | 81.9 |
| $G_3$-LoRA (L) | 93.1 | 84.3 | 87.7 | 82.4 | 87.8 | 86.2 | 83.1 | 54.3 | 82.3 |
| $G_3$-(P, L) | 92.8 | 84.1 | 88.7 | 82.6 | 88.2 | 86.2 | 83.3 | 54.7 | 82.6 |
| $G_3$-(A, P) | 93.1 | 83.8 | 89.1 | 82.3 | 88.1 | 85.8 | 82.6 | 55.1 | 82.5 |
| $G_3$-(A, L) | 92.7 | 84.5 | 88.4 | 82.8 | 88.2 | 86.1 | 83.5 | 54.6 | 82.6 |
| $G_3$-(A, P, L) | 92.8 | 84.6 | 88.1 | 82.5 | 87.7 | 85.5 | 83.2 | 53.8 | 82.3 |
| $G_3$-**(P, B, L)** | **93.6** | **84.9** | **89.3** | **83.1** | 88.2 | **86.5** | 83.9 | **55.8** | **83.2** |
| $G_3$-(A, P, B) | 93.3 | 83.9 | 88.5 | 82.2 | 88.4 | 86.2 | 83.5 | 55.3 | 82.6 |
| $G_3$-(A, B, L) | 93.4 | 84.2 | 88.9 | 82.6 | 87.8 | 85.8 | **84.2** | 54.9 | 82.7 |
| $G_3$-(A, P, B, L) | 92.2 | 84.4 | 88.7 | 82.3 | **88.5** | 86.2 | **84.2** | 54.2 | 82.5 |
| $G_4$-Adapter (A) | 92.8 | 85.2 | 89.1 | 83.5 | 87.8 | 86.5 | 84.2 | 56.3 | 83.2 |
| $G_4$-Prefix (P) | 92.8 | 84.6 | 89.5 | 82.6 | 87.4 | 86.5 | 83.8 | 55.8 | 82.8 |
| $G_4$-BitFit (B) | 93.8 | 84.9 | 89.5 | 83.3 | 88.7 | 86.8 | 84.4 | 55.2 | 83.3 |
| $G_4$-LoRA (L) | 93.3 | 84.7 | 89.3 | 82.7 | 88.3 | 86.2 | 82.7 | 54.7 | 82.7 |
| $G_4$-(P, L) | 93.8 | 85.3 | 89.6 | 83.6 | 88.6 | 86.8 | 84.6 | 56.3 | 83.5 |
| $G_4$-(A, P) | 93.8 | 84.9 | 89.8 | 84.3 | 88.5 | 86.6 | 84.8 | 56.7 | 83.6 |
| $G_4$-(A, L) | 93.7 | 85.6 | 89.5 | 84.1 | 88.2 | 86.6 | 85.2 | 55.4 | 83.5 |
| $G_4$-(A, P, L) | 94.2 | 85.2 | 89.6 | 83.9 | 88.2 | 86.4 | 84.9 | 55.9 | 83.5 |
| $G_4$-(P, B, L) | 93.8 | **85.9** | 89.8 | 83.6 | 88.6 | 86.9 | 85.2 | 56.3 | 83.7 |
| $G_4$-**(A, P, B)** | **94.4** | 85.7 | **90.1** | **84.8** | **88.9** | **87.2** | 85.3 | **57.3** | **84.2** |
| $G_4$-(A, B, L) | 93.8 | 85.3 | 89.5 | 84.1 | 88.8 | 86.7 | **85.5** | 56.6 | 83.7 |
| $G_4$-(A, P, B, L) | 94.1 | 85.4 | 89.7 | 84.4 | 88.5 | 86.5 | 85.2 | 56.8 | 83.8 |

Table 16: Average performances (low-compute, low-epoch regime: 100 random models, 3 tuning epochs) on the GLUE datasets using the T5-base pretrained backbone model. We compare adding different layer grouping constraints to the $\mathcal{S}_0$ design space. Layer grouping is based on 8 groups.

| Layer Grouping | SST-2 | MNLI | QNLI | QQP | RTE | STS-B | MRPC | CoLA | Avg |
|---|---|---|---|---|---|---|---|---|---|
| $\mathcal{S}_0$-models | 76.9 | 70.1 | 72.5 | 73.3 | 63.6 | 71.7 | 73.8 | 24.3 | 65.7 |
| Increasing | 83.2 | 74.1 | 76.6 | 77.1 | 67.7 | 76.8 | 74.7 | 30.0 | 70.0 |
| Uniform | 83.6 | 73.4 | 78.0 | 77.9 | 68.2 | 76.4 | 78.6 | 34.2 | 71.3 |
| Decreasing | 80.3 | 71.6 | 77.4 | 75.5 | 67.0 | 75.3 | 77.2 | 26.4 | 68.9 |
| **Spindle** | **86.2** | **74.3** | **79.1** | **78.6** | **68.5** | **77.4** | **79.5** | **35.1** | **72.3** |
| Bottleneck | 83.2 | 73.1 | 75.8 | 77.6 | 67.9 | 75.3 | 78.2 | 31.4 | 70.3 |

Table 17: Performances of different tuning methods on the SuperGLUE datasets using the XLNet-base (upper part) and XLNet-large (lower part) pretrained backbone models, respectively. The results are averaged over 10 random runs. The $\mathcal{S}_4$-model and $\mathcal{S}_4$-3b-model perform significantly better than the second-best PEFT methods in all the eight datasets at the significance level $p < 0.05$ (*) or even $p < 0.01$ (**).

| Method | BoolQ | CB | COPA | MultiRC | ReCoRD | RTE | WiC | WSC | Average |
|---|---|---|---|---|---|---|---|---|---|
| Adapter | 72.8 | 71.3/78.0 | 64.0 | 67.0/24.5 | 71.0/71.8 | 76.2 | 65.0 | 60.8 | 66.2 |
| Prefix | 72.0 | 70.5/77.0 | 63.3 | 66.4/23.8 | 69.9/71.0 | 75.5 | 64.4 | 60.8 | 65.9 |
| BitFit | 71.8 | 70.0/76.2 | 62.8 | 65.8/22.6 | 69.4/70.6 | 74.5 | 64.8 | 60.6 | 65.2 |
| LoRA | 72.2 | 71.1/77.8 | 64.7 | 67.4/24.8 | 70.8/71.3 | 76.8 | 65.1 | 61.1 | 66.4 |
| $\mathcal{S}_4$-model | **73.8**$^{**}$ | **71.7/78.4**$^{*}$ | **65.9**$^{**}$ | **68.2/25.5**$^{**}$ | **71.1/72.0**$^{*}$ | **78.4**$^{**}$ | **65.8**$^{*}$ | **62.6**$^{*}$ | **67.5** |
| Adapter | 74.4 | 71.4/81.1 | 67.4 | 68.8/26.4 | 71.7/72.4 | 80.8 | 68.0 | 64.6 | 68.8 |
| Prefix | 72.4 | 70.0/78.3 | 66.9 | 68.8/25.8 | 70.9/71.2 | 78.8 | 66.9 | 64.0 | 67.7 |
| BitFit | 71.1 | 70.7/79.8 | 68.0 | 68.6/25.4 | 71.1/71.6 | 80.4 | 67.2 | 64.3 | 68.1 |
| LoRA | 74.1 | 72.1/80.9 | 67.9 | 69.1/26.8 | 72.0/72.8 | 81.0 | 67.8 | 64.4 | 69.0 |
| $\mathcal{S}_4$-3b-model | **76.8**$^{**}$ | **74.6/81.9**$^{**}$ | **68.6**$^{**}$ | **69.5/27.1**$^{*}$ | **72.4/73.3**$^{*}$ | **81.2**$^{*}$ | **68.2**$^{**}$ | **64.8**$^{*}$ | **69.7** |

Table 18: Total training time (low-compute, low-epoch regime: 100 random models, 3 tuning epochs) on the GLUE datasets using the T5-base pretrained backbone model with 8 A100 GPUs from $\mathcal{S}_0$ to $\mathcal{S}_1$.

| SST-2 | MNLI | QNLI | QQP | RTE | STS-B | MRPC | CoLA |
|---|---|---|---|---|---|---|---|
| 18 mins | 22 mins | 20 mins | 40 mins | 8 mins | 12 mins | 8 mins | 6 mins |

