# OpenReview forum: "Parameter-Efficient Fine-Tuning Design Spaces"
_ICLR.cc/2023/Conference — ICLR 2023 poster_

### Official Review · Reviewer_CSJR · 2022-10-23

**Confidence:** 4
**Clarity, Quality, Novelty And Reproducibility:** 1. Mean and variance are required for…
**Correctness:** 3
**Technical Novelty And Significance:** 2
**Empirical Novelty And Significance:** 2
**Recommendation:** 5

**Strength And Weaknesses:**

Strength:

1. The proposed architecture beats traditional widely-used approaches, like LoRA.
2. The authors empirically explore the unified search space of parameter-efficient training.


Weaknesses:

My main concern comes from the application of the proposed methods: If a simple parameter-efficient training method (e.g., LoRA) can achieve comparable results, why do we need to design a heavy search algorithm to search for the best solution with only marginal improvements?


**Experiment Results**

1. Different architectures have different best settings. It is time-consuming if we need to do a grid search given a new architecture.

2. Table 5 shows that the improvements over LoRA mainly come from step 4. The authors should report more solid performance improvements to show the cost of searching is valuable.

3. GLUE contains several small-size datasets, like RTE, and CoLA. Different random seeds can contribute to over 5+ accuracy variance. As we can see, the improvements over baselines mainly come from these two datasets. Mean and variance are required for convincing results.


**Experiment Settings**

"we (i) randomly sample 100 models from the S0 design space that satisfy each grouping pattern constraint (Figure 2); (ii) fine-tune with 3 epochs; and (iii) compute the average performances for each design space. We will follow this procedure as we progressively add new constraints later."

The authors should report the costs of pre-training 100 models.

Furthermore, the authors should report the mean and variance across 100 models. If the gap between mean performance and the best performance is small, do we really need such search space?



**Experiment Conclusions**

The step2 and step3 seem to be artificial considering that the final decision is tuning all groups and uniformly allocating the number of trainable parameters.

If the authors can address these concerns, I'd like to raise the score.



**Summary Of The Paper:**

This paper empirically evaluates the best practice of parameter-efficient fine-tuning to achieve comparable performance with as few trainable parameters as possible. To achieve this goal, the authors design and implement different technique combinations and find the best solution via downstream experiments. Specifically, the pipeline contains four steps: 1) grouping layers into different sets; 2) adding trainable parameters towards each group; 3) deciding which group should be trained; 4) assigning groups with different training strategies.

Experiments show that spindle-style grouping, uniformly allocating trainable parameters to each layer, and tuning all layers achieve the best practice performance.  Step 4 requires the evaluation of different strategy combinations from Adapter (A), Prefix (P), BitFit (B), and LoRA (L).   Different architectures show different best combination settings.



**Summary Of The Review:**

The proposed combination method requires massive experiments and different architectures report different best settings. It is still unclear whether the proposed method is applicable to diverse architectures. Furthermore, the improvements over baselines are marginal in Table 5, and an ablation study is required to evaluate the contribution of different settings in the final performance.

---

> ### Author Response · Authors · 2022-11-19
> **Response to Reviewer CSJR**
>
> Thank you for the insightful comments, and being specific in questions for increasing your score.
>
>
> ### On the main concern about the gain vs. cost
>
> First, we do all the search process on T5 models, which result in models in our final parameter-efficient finetuning design space ($\mathcal{S}_4$ models). The $\mathcal{S}_4$ models perform significantly better than the second-best parameter-efficient fine-tuning methods in all the eight datasets at the significance level p<0.05 (or even p<0.01).  We have updated **Table 4**.
>
> Second, in the updated **Table 5**, we directly apply the design patterns discovered from T5 models to RoBERTa models **with no extra search cost**. It also shows that the $\mathcal{S}_4$ models perform significantly better than the second-best parameter-efficient fine-tuning methods in all the eight datasets at the significance level p<0.05 (or even p<0.01). This suggests that  our discovered patterns (on T5) could be directly generalized to other architectures (like RoBERTa) with no extra search cost and could consistently and significantly boost performance compared to existing methods.
>
>
> Third, we further perform new experiments by directly applying our searched patterns to XLNet (**without any extra search**) and evaluate the performances on SuperGLUE in the updated **Table 16 in the Appendix**. Again, the $\mathcal{S}_4$ models perform significantly better than the second-best parameter-efficient fine-tuning methods in all the eight datasets at the significance level p<0.05 (or even p<0.01). Note that the generalization of the discovered design patterns to more other architectures and tasks will further lead to lower **amortized cost** of our method due to its “searched once, applied everywhere” nature.
>
>
> Last but not least, we still want to highlight that the total training time cost (from the search process on T5 models) for every step is practical. For example, the computing time of every task (100 models) from our $\mathcal{S}_0$ space to $\mathcal{S}_1$ space is shown in **Table 18**.
>
>
> ### On the mean and variance of reported results
> We actually have reported the mean of multiple random runs in all the tables (for example,  the reported results are averaged over 100 sampled models in Table 1–3, the reported results are averaged over 20 random runs in Table 4 and Table 5). To provide more convincing results, we show the statistical significance between our methods and the second best baseline methods (paired t-test with p<0.01/0.05) in the updated **Table 4 and 5**.
>
>
> ### On the necessity of Step 2 and 3
>
> As for Step 2 (the allocation of the number of training parameters) and Step 3 (the tunable groups), they are also important design choices for parameter-efficient fine-tuning. Before actual searching, we can not confirm the optimal design choices for those factors. From our experiments, adding the discovered constraint for these two steps helps the performances, e.g., when applying uniform allocation patterns, $\mathcal{S}_2$ models are better than $\mathcal{S}_1$ models (74.0 vs. 73.3) by comparing the best performances in Table 2 and Table 1. Also after the adoption of tuning all the groups, $\mathcal{S}_3$ models are better than $\mathcal{S}_2$ models (74.9 vs. 74.0) by comparing the best performances in Table 3 and Table 2. These middle steps could also be viewed as ablation studies to show the necessity of every step.

---

> > ### Comment · Reviewer_CSJR · 2022-11-29
> > **Response**
> >
> > Thanks for the follow-up response and new experimental results.  The search costs in table 18 seem to be acceptable.
> >
> >
> > It is not surprising that the final performance is improved after a grid search. I still have concerns about the novelty of experimental findings. The authors say:
> >
> > "We emphasize that our goal is to demonstrate how the perspective of design spaces can help inform
> > PEFT research, rather than to find out the “best” design space or method. For computational efficiency, it is beyond the scope of this work to enumerate all possible constraints with respect to the design space components (Section 3).".
> >
> > If the target is to demonstrate how the perspective of design spaces can help inform PEFT research, I have no idea how step4 helps PEFT research since the final architecture is not "interpretable".
> >
> >
> > The new experiment results somehow address my main concerns, and I would like to raise my score from 3 to 5.

---

> > > ### Author Response · Authors · 2022-12-01
> > > **Thank you for your comments**
> > >
> > > Thank you for your detailed reply. We are so glad that the new experiments have addressed your main concerns. Here are our point-by-point responses:
> > >
> > > > "It is not surprising that the final performance is improved after a grid search. I still have concerns about the novelty of experimental findings"
> > >
> > > First, we would like to clarify that:
> > >
> > > * The PEFT design spaces are explored with *T5-base/3b* on *GLUE*.
> > > * The final performance is evaluated on *RoBERTa-base/large*, *BART-base/large*, *XLNet-base/large* on *summarization*, *machine translation*, and *eight SuperGLUE datasets*.
> > >
> > > Given that the experimental findings generalize with different backbones and different tasks, we humbly believe that they are interesting rather than "not surprising".
> > >
> > >
> > > Second, past PEFT strategies are "*equally* assigned to different pretrained layers" (see Paragraph 2 of Section 1). In sharp contrast, one of our experimental findings assigns proper tuning strategies to *different* layer groups (which generalizes to different backbones and different tasks), thus justifying its novelty.
> > >
> > >
> > > > "I have no idea how step4 helps PEFT research since the final architecture is not ’interpretable’"
> > >
> > > Different from past PEFT strategies that are "*equally* assigned to different pretrained layers" (see Paragraph 2 of Section 1), our Step 4 assigns proper tuning strategies to *different* groups: this is novel and interpretable.
> > >
> > > For example, we can also see from T5-base and T5-3b that Adapter and LoRA are the best strategies in lower groups, while BitFit is the best strategy in upper groups. As shown in the experiments, findings with T5-base/3b on GLUE can generalize to RoBERTa-base/large, BART-base/large and XLNet-base/large on summarization, machine translation, and eight SuperGLUE datasets. This shows promising potential impact to PEFT research.
> > >
> > > **Summary**
> > >
> > > We hope that the above could clarify on your concern about "the novelty of experimental findings" and why the PEFT design space may add value to the (probably huge) existing body of literatures on PEFT research, since design space for PEFT is still in its infancy. Again, thank you and the other reviewers for helping us make the paper better! We will acknowledge such help at the end of the revised paper.

---

> ### Author Response · Authors · 2022-11-29
> **Response to Reviewer CSJR**
>
> Dear Reviewer,
>
> We provided our response two weeks ago. Does it address your questions? We are more than happy to answer any further questions.
>
> Thanks!

---

### Official Review · Reviewer_3ina · 2022-10-24

**Confidence:** 4
**Correctness:** 2
**Technical Novelty And Significance:** 4
**Empirical Novelty And Significance:** 3
**Recommendation:** 8

**Clarity, Quality, Novelty And Reproducibility:**

Clarity: low
Quality: medium-to-high.
Novelty: high
Reproducibility: hard without the source code, although this is promised

**Strength And Weaknesses:**

STRENGTHS
1. A proposal to structure the different PET proposals into a unifying design space. In this sense, it is similar to He et al. (2022) (on this, a more thorough explanation of the main difference with that paper would be welcome)
2. A pragmatic way of searching over that space
3. Empirical results showing that the discovered design is better than previously proposed architectures, and generalize to new architecture and tasks. Specificailly, the obtained fine-tuning strategy carries over from T5 to Roberta, indicating that the findings are not ad-hoc for the T5 architecture. Similarly, the discovered design carries over from GLUE to a very different tasks (summarisation on XSUM, and translation for en-ro).

WEAKNESS
1. From the claims, I was expecting to read a "Evolved Transformer for parameter-efficient training" paper. This is not it, in that the search space is tree-based, and the design selection is greedy over each of the four stages. This is still interesting, as current proposals always felt wasteful in that they run the same architecture on all layers, but not necessarily what was expected from the abstract
2. Clarity. The paper is easy to read, but hard to understand in its details. For instance
 - what are the four groups? A very quick search of the provided reference (Dosovitskiy et al. (2020) does not answer that
 - the paper (specially the appendices) contain tons of tables and numbers, but it is not clear - even for a non-casual reader - what all those tables are saying. As an example, 4.3 seems to make the claim that the discovered design space carries over from T5-base to T5-3b, but it is not clear what table needs to be compared with what table. I compared Table 15 with all of Table 8-11 and from those it seems that this claim is actually not true (this is, the best strategy assignment per group for T5-3b is different than the best strategy assignment per group for T5-base)


**Summary Of The Paper:**

Parameter-efficient training (PET) is a very popular technique for domain and/or task adaptation of large pre-trained models. As opposed to in-context learning, it does perform weight updates, but much less so than full fine-tuning (0.5% of the total parameters in this case).
Many different techniques and architectures have been proposed over the last few years, and this paper proposes to:
1. model that design space in a more structured way
2. go over that design space empirically and discover possible new designs for PET

As a result they do not only show that this is possible, but also that the obtained design have some type of generalization, carrying over to different architectures and tasks

**Summary Of The Review:**

A good and rather exhaustive exploration of different design choices, addressing the issue that current PET strategies are rather rigid. The obtained design performs better than previous strategies.

---

> ### Author Response · Authors · 2022-11-19
> **Response to Reviewer 3ina**
>
> Thanks for your positive assessment and constructive feedback.
>
>
> ### On the claims in the abstract
>
>
> We have revised the abstract by adding more clarity of our method.
>
>
> ### On the clarity
>
> For the layer groups, they are subunits in large models where we view several consecutive layers as a whole. We are sorry that we made a typo for the reference, we have updated the correct one [1] in the updated paper.
>
> For the comparison between the conclusions drawn from T5-base and T5-3b, we described that in Section 4.3, where the patterns for layer grouping (Table 1 vs. Table 12), parameter allocation (Table 2 vs. Table 13) and tunable groups (Table 3 vs. 14) are consistent while the patterns for strategy assignments (Table 8-11 vs. Table 15) are a bit different due to the significant model size increase. We have added more references and descriptions to these tables in the updated paper.
>
> Reference: [1] 	Radosavovic, Ilija, et al. "Designing network design spaces." Proceedings of the IEEE/CVF conference on computer vision and pattern recognition. 2020.

---

### Official Review · Reviewer_CmEs · 2022-10-25

**Confidence:** 4
**Correctness:** 3
**Technical Novelty And Significance:** 3
**Empirical Novelty And Significance:** 3
**Recommendation:** 6

**Clarity, Quality, Novelty And Reproducibility:**

The paper is well written, easy to follow, and it's relatively easy to implement so that other researchers and practicioners can replicate the results (module some missing details, as indicated in the weaknesses above). The strategy is quite novel, altough a comparison against strong and competitive baselines (AutoML) is needed.

**Strength And Weaknesses:**

**Strengths**
- The paper is excellently written, it's easy to follow, and enough details are provided for practitioners and other researchers to implement the method.
- The proposed approach seems to provide quite consistent improvements (2 out of the 3 backbones improve results w.r.t. full fine-tuning).
- Although there are no experiments with runs using multiple random seeds, there are extensive experiments on multiple datasets (GLUE benchmark), and the best options are quite consistent across all datasets.

**Weaknesses**
- The main criticism is that the paper ignores other important baselines to choose where/how to tune the 0.5% of parameters during fine-tuning. For instance, the search space could be encoded in one of the many AutoML algorithms and let it find the optimal strategy, rather than relying on the fixed and hardcoded steps suggested by the paper, and then comparing the final results under a certain compute budget.
- The second shortcomming is that the three backbone architectures that have been tested are Transformer-based. The method should be ideally also tested with other backbones and/or domains (e.g. ResNets for image classification).
- The goal of having only 0.5% of tuneable parameters should be estated. For instance, if the goal is to reduce the fine-tuning cost, notice that the computational savings of tuning only 0.5% of parameters are nothing compared to the large amount of tuning needed to find the optimal strategy.
- It's not clear how the "Prefix" strategy can be applied independently on each group of layers. After all, it adds a new trainable token to the sequence. So, when "Prefix" it's added to a given layer (or group), all subsequent layers will have an additional token. No? Could you clarify this?
- The paper states that due to computational limitations, not all combinations were explored in this work. That's fine, but then the paper should be explicitly mention the cost of applying the proposed strategy w.r.t. the number of layers in the network and strategies considered.
Overall, the subspace explored in each design step by the paper isn't clear.
- The overall cost of finding a good strategy to find a small subset of parameters to fine-tune is very expensive, though. Thus, this reviewer has doubts that this will be applied in practice, in comparison to simply fine-tuning the entire network or just using Adapters or LoRa in all layers, which give competitive results with a much simpler (and cheaper) process.
- For example: when exploring the space $S_1$, depending on the number of layers, there are multiple options for the Increasing, Decreasing, Spindle, and Bottleneck options (e.g. for a network with 12 layers, with increasing strategy, valid configurations are $N_1 = 1, N_2 = 2, N_3 = 4, N_4 = 5$ or $N_1 = 1, N_2 = 2, N_3 = 3, N_4 = 6$), but it's not indicated which ones are recommended or were explored in the paper.
- In Table 6 (BART results), the bolded results should be those of "full" and not the proposed method, since "full" is better than the proposed method.
- Clarification: when multiple strategies are applied on the same group, it means that all the strategies are applied in all layers of the group, right? (e.g. $G_1-(A, L)$ means that in every layer of $G_1$ adapters are added and LoRa is applied).
- Suggestion: In table 7 (appendix A), add an extra (multirow) column indicating the number of epochs on each group, for better readability.

**Summary Of The Paper:**

The paper proposes a strategy to find a parameter-efficient tuneable architecture from a given pre-trained backbone neural network. In particular, the strategy consits of four phases exploring 1) how to group layers in the backbone neural network; 2) how to allocate the tuneable parameters within each group; 3) how to decide which groups to tune and which groups to keep frozen; 4) assign the best strategy to increase the parameters in each group. The paper uses this strategy to find an architecture based on BERT, RoBERTa and BART that tunes only 0.5% of the original parameters, and achieves better results than full fine-tuning on GLUE for BERT and RoBERTa backbones.

**Summary Of The Review:**

The paper proposes a simple-to-implement strategy to find a good way of tune a small set of parameters instead of fine-tuning the entire network, which gives excellent results in the GLUE benchmarks for different Transformer-based backbone networks. The work, however, has several and important drawbacks mentioned above, thus I think it's marginally below the acceptance threshold. If the authors are able to satisfactorily address some of my concerns, I'll be happy to increase my score.

UPDATE AFTER DISCUSSION:
The authors have addressed some of my concerns and clarified my questions. Thus, I'm increasing the score from 5 to 6. I thank the authors for their detailed response.

---

> ### Author Response · Authors · 2022-11-19
> **Response to Reviewer CmEs**
>
> hank you for the insightful comments, and being specific in questions for increasing your score.
> ### On the selection of searching strategy
> The reason why we are not considering AutoML methods like NAS is that such methods generally find a single network instance tuned to a specific setting, which lacks interpretation and the discovery of the parameter-efficient fine-tuning design patterns [1]. Our designed space and discovery steps can help the understanding of the parameter-efficient fine-tuning and discover the design patterns that allow generalization to different settings.
>
> ### On the selection of backbone models
> We mainly test with four different Transformer-based models (T5, RoBERTa, BART and the newly added XLNet) rather than other backbones like ResNets because we focus on NLP domains. In fact, the ICLR paper “Towards a Unified View of Parameter-Efficient Transfer Learning” also focuses on Transformer-based models with NLP applications. Nonetheless, we are open to changing the title (Parameter-Efficient Fine-tuning Design Spaces for NLP) if that will add more clarity.
>
> ### On the goal of tuning 0.5% parameters
> By tuning 0.5% parameters, the major saving comes from the storage cost when applying the large model into complex tasks. As discussed in our Introduction, if we finetune all the parameters, we need to store different models for different tasks (e.g., $175$B parameters for GPT-$3$). This would make it difficult to deploy in real-world NLP systems. Thus we need PEFT methods to tune only a small portion of the parameters.
>
> ### On the clarification of prefix
> The way we are using prefix is that we append the trainable vectors to the beginning of the hidden sequence if a certain layer use prefix. And the trainable vectors only help modify the hidden representation of the sentence in the current layer. They will not be passed to later layers. So when prefix is added to a given layer, the subsequent layers will not have these additional vectors.
>
> ### On the size of the subspace
> The total subspace is large for every design choice, especially when there are more layers in larger models. For a 12 layer transformer model with 125M trainable parameters, there are around $ C_{12}^4 C_{62500}^4 2^4 (C_4^1+C_4^2 + C_4^3 + C_4^4)^4  = 10^{22} $ possible configuration in the $S_0$ design space. After the first search, the possible configuration is lower down to  $C_{62500}^4 2^4 (C_4^1+C_4^2 + C_4^3 + C_4^4)^4  = 10^{10}$.
>
> ### On the performance gains vs. cost:
> First, we do all the search process on T5 models, which result in models in our final parameter-efficient finetuning design space ($\mathcal{S}_4$ models). The $\mathcal{S}_4$ model performs significantly better than the second-best parameter-efficient fine-tuning method in all the eight datasets at the significance level p<0.05 (or even p<0.01).  We have updated **Table 4**.
>
> Second, in the updated **Table 5**, we directly apply the design patterns discovered from T5 models to RoBERTa models **with no extra search cost**. It also shows that the $\mathcal{S}_4$ model performs significantly better than the second-best parameter-efficient fine-tuning method in all the eight datasets at the significance level p<0.05 (or even p<0.01). This suggests that our discovered patterns (on T5) could be directly generalized to other architectures (like RoBERTa) with no extra search cost and could consistently boost performance compared to existing methods.
>
> Third, we further perform new experiments by directly applying our searched patterns to XLNet (**without any extra search**) and evaluate the performances on SuperGLUE in the updated **Table 17 in the Appendix**. Again, the $\mathcal{S}_4$ model performs significantly better than the second-best parameter-efficient fine-tuning method in all the eight datasets at the significance level p<0.05 (or even p<0.01). Note that the generalization of the discovered design patterns to more other architectures and tasks will further lead to lower **amortized** cost of our method due to its “*searched once, applied everywhere*” nature.
>
> Last, we show the total training time of every task (100 models) from our $\mathcal{S}_0$ space to $\mathcal{S}_1$ space (5 patterns) as a reference in **Table 18**. The training time is practical.
>
> ### On the other clarification questions
> * For the spindle grouping pattern, the 2, 4, 4, 2 works the best.
> * In our setup, the performances of full tuning are not compared to parameter-efficient methods for fair comparisons. The performances are mainly for references about the “upper bound" of the tasks. We have updated all the tables to avoid confusion.
> * We apply all the strategies to all layers in the groups when multiple strategies are applied on the same group.
> * We have followed your suggestion and updated **Table 7**
>
> Reference:
> [1] Radosavovic, Ilija, et al. "Designing network design spaces." Proceedings of the IEEE/CVF conference on computer vision and pattern recognition. 2020

---

> > ### Comment · Reviewer_CmEs · 2022-11-20
> > **Thanks for your response**
> >
> > I thank the authors very much for their very detailed responses to my quesions and concerns.
> >
> > My main criticism is still valid. The authors argue that they didn't compare against AutoML methods because they "find a single network instance tuned to a specific setting, which lacks interpretation and the discovery of the parameter-efficient fine-tuning design patterns". However, the fact that the architecture found by these methods is not "interpretable" is not a good enough reason (in this reviewer opinion) to exclude the method from the baselines. This still remains an important shortcoming of the paper.
> >
> > The authors have, nevertheless, addressed many other of my questions in their rebuttal, and for that I'm going to slightly increase my score.
> >
> > PS: Given that the paper doesn't evaluate the algorithm in other tasks other than NLP, I do recommend the authors limiting the scope of the title (e.g. adding a "for NLP" suffix as they suggest).

---

> > > ### Author Response · Authors · 2022-11-21
> > > **Thanks**
> > >
> > > Thanks for the encouraging feedback. We agree with your opinion that the design space can be encoded in AutoML. That's actually a great point: we are working on integrating our discovered design patterns (anonymous link: https://github.com/anonymous-sc/parameter-efficient-fine-tuning-design-spaces) into an [anonymous] open-source AutoML library (developed by authors) that is popular with both academia and industry. For example, users can specify "grouping pattern=spindle" in config.ini to enjoy parameter-efficient fine-tuning performance boosts without additional searching costs. From users' perspective, all it takes is just a few lines of code.
> > >
> > > Now back to the scope of this work. Although we do develop popular AutoML tools, however, in the domain of parameter-efficient fine-tuning, we still choose to focus on the design spaces, rather than exploring all possible AutoML algorithms. There are three major reasons for this decision.
> > >
> > > First, the study of design spaces has already been well motivated and established in
> > >
> > > [1] Ilija Radosavovic, Raj Prateek Kosaraju, Ross Girshick, Kaiming He, Piotr Dollar. "Designing network design spaces." CVPR 2020.
> > >
> > > [2] Jiaxuan You, Rex Ying, Jure Leskovec. "Design Space for Graph Neural Networks." NeurIPS 2020.
> > >
> > > For example, Section 1 of [1] says,
> > >
> > > > "Despite the effectiveness of NAS, the paradigm has limitations. The outcome of the search is a single network instance tuned to a specific setting (e.g., hardware platform). This is sufficient in some cases; however, it does not enable discovery of network design principles that deepen our understanding and allow us to generalize to new settings. In particular, our aim is to find simple models that are easy to understand, build upon, and generalize."
> > >
> > > From the past years' experience of growing our AutoML library, we'd like to share that building solutions that are "easy to understand" (one limitation of NAS) and "generalizable" (e.g., generalizable to RoBERTa, BART, XLNet, summarization, machine translation, and eight SuperGLUE datasets as in our design space work) has high practical values.
> > >
> > > Second, neither [1] nor [2] is based on AutoML algorithms such as NAS. More concretely, the "low-compute, low-epoch training regime" (for exploring design spaces) in our work is almost the same as that in [1] (the second paragraph of Section 3.1 in [1]).
> > >
> > > Third, although design space is not new, design space for parameter-efficient fine-tuning is still novel and important. As with any new paradigms, it is expected that there is still space for improvements. Although at this moment there is no existing work on NAS for parameter-efficient fine-tuning, we think expanding this work to NAS should be regarded as one of the top priority topics to be dealt with in our future work. If you think that it is necessary, we would like to include more discussions or concluding thoughts in the paper to inspire future research.
> > >
> > > PS: We will add a "for NLP" suffix in the title.
> > >
> > > Again, thank you and the other reviewers for helping us make the paper better! We will acknowledge such help at the end of the revised paper.

---

### Official Review · Reviewer_8TXz · 2022-10-25

**Confidence:** 4
**Correctness:** 3
**Technical Novelty And Significance:** 2
**Empirical Novelty And Significance:** 2
**Recommendation:** 6

**Clarity, Quality, Novelty And Reproducibility:**

### Clarity and quality

The paper is generally well-written, the few remaining typos do not affect clarity. I have no complaints about its presentation quality.

Typo: [tables 9 and 10, caption] -- “ G1-(L, A) “ - shouldn’t it be (A, L)?

### Novelty

To the best of my knowledge, (1) the specific optimal combinations of design choices is novel and (2) the claim that these design choices generalize between different models, if properly verified, is also novel.

### Reproducibility

The current version of the paper is difficult to reproduce, since (1) the source code is not available, (2) the detailed experiment setups are also missing. To the best of my knowledge, even basic settings such as optimizer hyperparameters / batch size / learning rate schedule are missing. Since this is a practical paper, both code and configuration are necessary: not having them would undermine the core contribution of the paper.

**Strength And Weaknesses:**

### Strengths

To the best of my understanding, this paper is a glorified hyperparameter search. [If authors disagree, I will eagerly hear their argument] However, this is not a bad thing: parameter-efficient finetuning (arguably) badly needs a systematic hyperarameter search. The area is flooding with various algorithms and they do not always compare in the same setting. To that end,

- the paper finds specific optimal combinations of algorithms for popular models. Even if we ignore generalization, these combinations are immediately useful to practitioners.
- if authors truly found a combination of layers that generalizes across models (see a few concerns below), this combination could have an even greater practical impact, since it would be a great default combination for new transformer-based models
- the paper reports several negative results in Section 4.2.3, 4.2.4. These are (again, arguably) not surprising, but still useful to save the time of researchers that tune their PEFT algorithms.


### Weaknesses

All my concerns are about unjustified or under-explored design decisions:

1. Using 4 groups: while authors mention that it is justified by some vision model, it is unclear why this choice is justified for the NLP tasks in question. One way to validate this choice would be to run an ablation analysis: if doubling the number of groups does not change results, then 4 groups are likely enough.
2. The proposed search protocol answers one question at a time, e.g. group sizes first, then algorithms. To the best of my understanding, the order of these design spaces is arbitrary. Do the results hold if authors change the order in which they apply design constraints? e.g. select methods first, then determine group sizes.
3. Authors explore how to allocate parameters between layer groups. However, when a single parameter group uses multiple methods (e.g. lora and soft prompts), how do they allocate parameters between groups? E.g. more prompts or higher adapter rank?


### Questions

1. To the best of my knowledge, T5 and BART are encoder-decoder models, while RoBERTa is encoder-only. How do authors apply grouping in encoder-decoder models? e.g. does the spindle grouping apply encoder and decoder separately, or pretend that they are stacked together? Does LoRA apply to T5/BART's cross-attention, or only self attention? Does any of this affect model quality?

2. Concerning the en-ro machine translation experiments: to the best of my understanding, the pre-trained models were not trained on the destination language. Furthermore, its tokenizer / embeddings might be unsuitable for this language. How exactly do you run these experiments?

3. the search protocol for finding optimal design spaces is based on random search. The same problem can be solved with [A] hyperparameter optimization techniques, e.g. TPE or gaussian process optimization (see Ray-tune or Optuna) or [B] neural architecture search. Why did authors opt for random search?

**Summary Of The Paper:**

This paper systematically looks for the best way to combine parameter-efficient fine-tuning methods. Authors consider 4 design decisions: (1) how to group layers together (that layers within one group are treated equally), (2) how to allocate the "budget" of trainable parameters between groups, (3) whether to tune each of the groups and (4) the combination of PEFT algorithms to use in each group. In (2) and (3), authors find that naive solution is optimal. In (1) and (4), they find a combination of methods that outperforms all individual strategies. Finally, authors demonstrate that this combination can generalize to different models (e.g. from T5 to RoBERTa, BART) -- and that it is fairly consistent among tasks.

**Summary Of The Review:**

To reiterate: I believe that this is paper is simply a systematic hyperparameter search -- in an area that could use just such a search. In its current form, the paper lacks several important details: justification of some choices (e.g. number of groups), verification of the search protocol (e.g. is it robust to the order of design spaces?) and reproducibility details (code, setups, config).

---

> ### Author Response · Authors · 2022-11-19
> **Response to Reviewer 8TXz**
>
> Thanks for your insightful comments. We humbly think that some concerns might be caused by misunderstanding, which we will explain in detail below. We hope that our response can clarify the misunderstandings and you can consider our work more favorably.
>
>
> ### On the justification of using 4 groups
>
> We are inspired by [1] and follow the idea of grouping the transformer layers into 4 groups. Nonetheless, we have followed your advice by performing experiments (**Table 16 in the Appendix**)  to validate the choices where we use 8 groups in the first searching step and observe consistent conclusions (the spindle patterns still work the best).
>
>
> ### On the discovery sequence
> We use the current searching orders because (we have revised our paper by adding these discussions in **Appendix C**):
>
> * To explore and understand the design patterns in all the layers in large pre-trained models in scale, it is necessary and more efficient to study the layers in the unit of groups. So we start with the grouping patterns.
>
> * Once figuring out the optimal grouping patterns, it is then important to explore how to allocate the trainable parameters to these different groups in order to study more subtle designs with fair comparisons (e.g., this would allow comparing different patterns of strategy assignments without the impact from different trainable parameters.).
>
> * Next, it becomes influential to examine which groups need to be learned during fine-tuning before we dig into the strategy assignment patterns. Because it is only meaningful to study assigning strategies to different groups after we figure out which groups need to be learned.
>
> * Finally, we study the tuning strategy assignment, which is the most subtle design.
>
>
> ### On the clarification about the parameter allocation
>
> We first allocate the training parameters **among different groups**. In every group, we then uniformly distribute the training parameters in one single group to **different tuning methods** if  there are multiple methods in one single group.
>
>
> ### On the clarification about the experiment settings
>
> During experiments, we use the same design choice for the encoder and decoder transformer layers. e.g, we apply the same design choice to the first encoder layer group and the first decoder layer group. Also, the spindle grouping patterns are applied to the encoder and decoder separately. And LoRA are applied to both self-attention and cross-attention in T5/BART’s decoders. The number of training parameters is also uniformly allocated between self-attention and cross-attention.
>
> For the BART on en-ro experiments, we used the mBART version to perform the experiments.
>
>
> ### On the choice of the searching protocol
>
> The study of design spaces based on the current searching protocol has already been motivated and established in prior works: [1] Ilija Radosavovic, Raj Prateek Kosaraju, Ross Girshick, Kaiming He, Piotr Dollar. "Designing network design spaces." CVPR 2020. [2] Jiaxuan You, Rex Ying, Jure Leskovec. "Design Space for Graph Neural Networks." NeurIPS 2020. For example, Section 1 of [1] says, "Despite the effectiveness of NAS, the paradigm has limitations. The outcome of the search is a single network instance tuned to a specific setting (e.g., hardware platform). This is sufficient in some cases; however, it does not enable discovery of network design principles that deepen our understanding and allow us to generalize to new settings. In particular, our aim is to find simple models that are easy to understand, build upon, and generalize." In fact, neither [1] nor [2] is based on HPO/NAS. More concretely, the "low-compute, low-epoch training regime" (for exploring design spaces) in our work is almost the same as that in [1] (the second paragraph of Section 3.1 in [1]).
>
>
> ### On the detailed experiment setup
> For the detailed experiment setting like the optimizer hyperparameters / batch size / learning rate, we report them in the Section 5.1. We have also uploaded source code to an anonymous GitHub repository: https://github.com/anonymous-sc/parameter-efficient-fine-tuning-design-spaces.
>
>
> Reference: [1] 	Radosavovic, Ilija, et al. "Designing network design spaces." Proceedings of the IEEE/CVF conference on computer vision and pattern recognition. 2020.

---

> > ### Author Response · Authors · 2022-11-29
> > **Response to Reviewer 8TXz**
> >
> > Dear Reviewer,
> >
> > We provided our response two weeks ago. Does it address your questions? We are more than happy to answer any further questions.
> >
> > Thanks!

---

> ### Author Response · Authors · 2022-12-03
> **Kind Reminder on Your Questions and Our Response with Provided Source Code**
>
> Dear Reviewer,
>
> Thanks again for your efforts in reviewing our paper, especially raising concrete questions where you expect our response.
>
> We have addressed all your questions in detail and provided source code. Would you mind acknowledging our rebuttal? As the discussion due is approaching, if you have any questions, let's discuss on the OpenReview system.
>
> Best,
> Authors

---

### Author Response · Authors · 2022-11-19
**General Response to All the Reviewers**

We want to thank all the reviewers for the positive assessment, insightful comments, and being specific in questions for increasing their scores. We have revised our paper accordingly and provided individual responses to each reviewer. The main changes to the paper are summarized as follows:

* Adding more experimental results and discussions, including:

    * Performing experiments suggested by Reviewer 8TXz to verify the design choice which grouping transformer layers into 4 groups and adding the **Table 16**.

    * Performing more random runs (20 runs), running significant tests (paired t-test) to verify the significance of our methods over the second best parameter-efficient fine-tuning methods and updating **Table 4** and **Table 5**.

    * Performing new experiments with **XLNet** backbone models evaluated on all the 8 datasets in **SuperGLUE** in **Table 17** to provide further evidence on the generalization of our discovered patterns.

* Describing the experimental setup and uploading source code to an anonymous GitHub repository: https://github.com/anonymous-sc/parameter-efficient-fine-tuning-design-spaces

* Adding discussions about the time cost of the discovery process in **Table 18** and describing the intuition behind the discovery orders in the **Appendix C**.

---

### Decision · Program_Chairs · 2023-01-20

**Decision:**

Accept: poster

**Justification For Why Not Higher Score:**

Low novelty from a methodological standpoint.

**Justification For Why Not Lower Score:**

After discussion, the reviewers (and the AC) felt that the empirical contribution of the paper was "enough" to warrant acceptance.

**Metareview: Summary, Strengths And Weaknesses:**

This paper studies how existing parameter-efficient finetuning methods can be combined to enable even greater parameter efficiency in pretrained language models. The main contribution of this paper is the showing that existing methods can be combined and can even generalize across different pretrained models. The main weakness is novelty: in some sense, the authors are performing hyperparameter tuning across the different methods.

**Note From Pc:**

if the above contains the word "oral" or "spotlight" please see: "oral" presentation means -> notable-top-5% and "spotlight" means -> notable-top-25%. As stated in our emails, we are disassociating presentation type from AC recommendations

**Summary Of Ac-Reviewer Meeting:**

We (CmEs, 3ina and the AC) had a Zoom discussion regarding the paper, and the consensus is that while the novelty is low (as reviewer 8TXz notes, in some sense this just glorified hyperparameter search), the empirical results (especially the fact the approach seems to generalize to other architectures and generation tasks) may be strong enough to warrant acceptance. However, I will note that the two reviewers who were less positive about the paper did not respond to the AC-reviewer meeting request.